

# The role of highly oxidized multifunctional organic molecules for the growth of new particles over the boreal forest region

Emilie Öström[1,2], Pontus Roldin[1,3], Guy Schurgers[4], Mikhail Mishurov[5], Zhou Putian[3] Niku Kivekäs[6], Heikki Lihavainen[6], Mikael Ehn[3], Matti P. Rissanen[3], Theo Kurtén[7], Michael Boy[3], Erik Swietlicki[1]

[1]Division of Nuclear Physics, Lund University, Lund, P.O. Box 118, 221 00, Sweden
[2]Centre for Environmental and Climate Research, Lund University, Lund, P.O. Box 118, 221 00, Sweden
[3]Department of Physics, University of Helsinki, Helsinki, Box 64, 00014, Finland
[4]Department of Geosciences and Natural Resource Management, University of Copenhagen, Copenhagen, 1350, Denmark
[5]Department of Physical Geography and Ecosystem Science, Lund University, Lund, 223 62, Sweden
[6]Finnish Meteorological Institute, Helsinki, Box 503, 00101, Finland
[7]Department of Chemistry, University of Helsinki, Helsinki, P.O. Box 55, 00014, Finland

*Correspondence to*: Emilie Öström (emilie.ostrom@nuclear.lu.se)

**Abstract.** Secondary organic aerosol particles (SOA) are important climate forcers, especially in otherwise clean environments such as the boreal forest. There are, however, major uncertainties in the mechanisms behind the formation of SOA, and in order to predict the growth and abundance of SOA at different conditions, process-based understanding is needed. In this study, the processes behind new particle formation (NPF) events and subsequent growth of these particles in the northern Europe sub-Arctic forest region are explored with the one-dimensional column trajectory model ADCHEM. The results from the model are compared with particle number size distribution measurements from Pallas Atmosphere-Ecosystem Supersite in Northern Finland. The model was able to reproduce the observed growth of the newly formed particles if a small fraction of the emitted monoterpenes that are oxidized by $O_3$ and OH undergo autoxidation and form highly oxidized multifunctional organic molecules (HOMs) with low or extremely low volatility. The modeled particles originating from the NPF events (diameter < 100 nm) are composed predominantly of HOMs. While the model seems to capture the growth of the newly formed particles between 1.5 and ~ 20 nm in diameter, it underestimated the particle growth between ~20 and 80 nm in diameter. Due to the high fraction of HOMs in the particle phase, the oxygen-to-carbon (O : C) atomic ratio of the SOA was nearly 1. This unusually high O : C and the discrepancy between the modeled and observed particle growth might be explained by the fact that the model did not consider any particle-phase reactions involving semi-volatile organic compounds with relatively low O : C. According to the model the phase state of the SOA (assumed either liquid or amorphous solid) had an insignificant effect on the evolution of the particle number size distribution during the NPF events. The results were sensitive to the method used to estimate the vapor pressures of the HOMs. If the HOMs were assumed to be extremely low volatile organic compounds (ELVOCs) or non-volatile the modeled particle growth was substantially higher than when the vapor pressures of the HOMs were estimated based on continuum solvent model



calculations using quantum chemical data. Overall, the model was able to capture the main features of the observed formation and growth rates during the studied NPF-events if the HOM mechanism was included.

# 1 Introduction

Atmospheric particles affect climate by scattering and absorbing solar radiation and by influencing cloud formation and
cloud optical properties. Their climate effect depends on both the size and composition of the particles and remains one of the largest uncertainties in global climate predictions (IPCC, 2013). Small-scaled, process-based models are important tools for studying different mechanisms behind aerosol formation and growth. It is crucial to understand these processes in order to improve the predictability of next generation climate and weather forecast models.

In this study, the growth of biogenic secondary organic aerosols (BSOA) over the boreal forest in northern Europe is
modeled and the results are compared to particle number size distribution measurements. New particle formation (NPF) events in boreal forests are frequent (Asmi et al., 2011; Kulmala et al., 2001; Tunved et al., 2003) and the newly formed particles can grow by condensation to the climate-relevant cloud condensation nuclei (CCN) size range, which starts at a diameter of ~50 nm (Kerminen et al., 2012). Komppula et al. (2005) found that particles in the boreal region in northern Finland are typically able to activate into cloud droplets when they reach diameters larger than 80 nm (the minimum
activation diameter varied from 50 to 128 nm). In boreal forests, the growth of the particles is dominated by condensation of organic compounds formed from oxidation of biogenic volatile organic compounds (BVOC) emitted by the vegetation (Kulmala et al., 2013). Studies have shown that NPF can provide a significant amount of CCN and thereby have a substantial climate impact (e.g. Jokinen et al., 2015; Kerminen et al., 2012; Merikanto et al., 2009; Scott et al., 2014; Spracklen et al., 2008).

The different ways to model the formation of BSOA found in the literature reflect the uncertainties of the formation mechanisms and also the often unknown properties of the condensable vapors. In many studies (e.g. Bergström et al., 2012; Farina et al., 2010; Fountoukis et al., 2014; Hodzic et al., 2009; Lane et al., 2008; Murphy et al., 2012) the vapors are assumed to be semi-volatile and in equilibrium with the (liquid, well-mixed) particles, making it possible to model formation of BSOA by simple gas-particle equilibrium partitioning (Pankow, 1994). In other studies (e.g. Scott et al., 2015; Spracklen
et al., 2008; Tunved et al., 2010; Westervelt et al., 2013) the vapors are assumed to be non-volatile and the irreversible particle growth is only limited by the collision rate between the vapor molecules and the particles. Recently, large-scale model studies (Jokinen et al., 2015; Langmann et al., 2014; Li et al., 2015; Riipinen et al., 2011; Yu, 2011) have included both mechanisms to be able to treat semi-volatile and non-volatile vapors, which have yielded a better agreement between model results and observations. This hybrid SOA formation mechanism is an important step forward. However, in order to
explicitly simulate the size-resolved condensational growth, models need to take into account how the chemical composition and curvature (Kelvin) effect vary with the size of the particles. Smog-chamber studies have often focused on the SOA formation from semi-volatile organic compounds (SVOCs). Recently the presence of highly oxidized multifunctional



organic molecules (HOMs) in the gas-phase has been shown in both lab and field studies (Ehn et al., 2014; Jokinen et al., 2015). Many HOMs can be low-volatility organic compounds (LVOCs) or even extremely low-volatility organic compounds (ELVOCs), while others are SVOCs (Kurtén et al., 2016). The volatility distribution and aging of SOA in models will significantly affect the model results of SOA formation (Hermansson et al., 2014). Furthermore, the phase-state of the

particles can affect the dynamics of the growth (Zaveri et al., 2014). Traditionally, SOA particles are assumed to be well-mixed liquids; however, recent experimental studies indicate that they can be solid-like at ambient conditions (Renbaum-Wolff et al., 2013; Saukko et al., 2012; Vaden et al., 2011; Virtanen et al., 2010), which may influence their growth and lifetime (with respect to evaporation) in the atmosphere (Roldin et al., 2014).

In this study, we assume the organic vapors to condense dynamically on the Fuchs-corrected surface area of the particles.

The two extremes of particle phase state are tested; either the particles are assumed to be well-mixed liquid droplets or they are assumed to be solid-like without diffusion in the particle phase and with the gas-particle partitioning being controlled by the composition at the surface. Based on Ehn et al. (2014), a formation pathway of HOMs by the oxidation of α-pinene, β-pinene and limonene is added. The aerosol dynamics are modeled along air-mass trajectories with an updated version of the Aerosol Dynamics, gas and particle phase CHEMistry and radiative transfer model (ADCHEM) (Roldin et al., 2011b). The

modeled results are compared to particle number size distribution measurement at the subarctic station in Pallas, northern Finland.

The aim is to constrain the contribution of HOMs to the activation and growth of new particles over the boreal forest region. The model approach is described in Sect. 2, followed by results and discussion in Sect. 3 and conclusions in Sect 4.

## 2 Method

ADCHEM was used to model the concentrations of gases and particles along air-mass trajectories ending at the Pallas Atmosphere-Ecosystem Supersite (67.97º N, 24.12º E, 565 m.a.s.l) (Lohila et al., 2015) in northern Finland. The emissions of different primary particulate and gaseous chemical species along the trajectories were derived from emission databases listed in Sect. 2.2. The modeled particle number size distributions for the Pallas site were compared to measured ones. The measurements were conducted with a differential mobility particle sizer (DMPS) covering dry particle mobility range 7 –

500 nm (Komppula et al., 2003). The instrument was connected to a non-standard inlet having cut-off diameter of approximately 5 μm (Lohila et al., 2015).

### 2.1 Air-mass trajectories

Based on the particle number size distribution data measured at Pallas between 2005 and 2010, days with NPF events suitable for modeling SOA formation were selected for detailed analysis. This included days with strong new particle

formation and subsequent growth of the new particle mode for at least 12 hours. This selection is roughly in line with type 1a events as defined by Dal Maso et al. (2005). The corresponding air-mass trajectories for these days were determined using





the Hybrid Single Particle Lagrangian Integrated Trajectory Model (HYSPLIT) (Draxler and Rolph, 2013) with meteorological data from the Global Data Assimilation System (GDAS), downloaded from NOAA Air Resource Laboratory Real-time Environmental Application and Display sYstem (READY) (Rolph, 2016). The trajectory resolution was linearly interpolated from 3 h to 1 min (the main model time step used in the simulations). The air-mass trajectories were calculated 7

days backward in time and ending at Pallas at 00, 03, 06, 09, 12, 15, 18 and 21 UTC. The analyzed cases were further decreased by including only those where all air-mass trajectories originated from clean marine environments. For each chosen new particle formation event the particle and gas-phase evolution along the air-mass trajectories were modeled. In 7 out of the in total 10 selected cases, the growth of the newly formed particle mode could be observed also at the day after the start of the event. For these cases we ran ADCHEM also for the day after the NPF event. In total the model was run along

136 air mass trajectories. Figure 1 shows the mean trajectories for each new particle formation and growth event.

Information on land-use along the trajectories was retrieved from the Global Land Cover Map for the Year 2000, GLC2000 database, European Commission Joint Research Centre (http://forobs.jrc.ec.europa.eu/products/glc2000/products.php). Land-use categories were used to calculate the dry deposition of gases and particles.

## 2.2 Emissions along the trajectory

All emissions were added at each model time step to the model layer closest to surface, where they were assumed to be instantaneously well-mixed within this layer.

### 2.2.1 Gas emissions

Anthropogenic emissions of CO, $NH_3$, non-methane volatile organic compounds (NMVOCs) (represented by 25 species (see table S1)), $NO_x$ and $SO_2$ were retrieved from the EMEP (European Monitoring and Evaluation Programme) database

(EMEP/CEIP 2014 Present state of emissions as used in EMEP models; http://www.ceip.at/webdab_emepdatabase/emissions_emepmodels/). Dimethyl sulfide (DMS) emissions from marine plankton were also retrieved from EMEP. The modeled $SO_2$ concentration in the surface layer 24 hours upwind from Pallas was nudged towards the measured $SO_2$ concentration at the station by increasing the emission of the gas when the modeled concentration was below the measured one. This applied to ~50 % of the studied trajectories, and for these cases the $SO_2$

concentration was increased by a median factor of 1.8. Nudging was done in order to get a more realistic nucleation rate since the modeled nucleation rate depends on the concentration of $H_2SO_4$ (Eq. 2 and 3 in Sect. 2.3), which is formed by the reaction between $SO_2$ and OH. The median modeled (with nudging) and measured $SO_2$ gas-phase concentrations during the NPF-events are shown in Fig. S1b.

Biogenic emissions (α-pinene, β-pinene, limonene, carene and isoprene) were estimated with the dynamic vegetation model

LPJ-GUESS (Smith et al., 2014), which simulates the carbon and nitrogen cycling in terrestrial vegetation and which contains algorithms for isoprene (Arneth et al., 2007) and monoterpene (Schurgers et al., 2009a) production and emission by plants. Vegetation is represented with plant functional types (PFTs), and we applied 11 tree species common for Northern



Europe, one generic shrub type and one herbaceous type (Table S2), applying the bioclimatic limits as in Hickler et al. (2012) and Schurgers et al. (2009b). The parameterization of the PFTs and their isoprene and monoterpene characteristics follows Schurgers et al. (2009b), but the monoterpene emissions were split into three separate sets (α-pinene, β-pinene and limonene), as well as a generic set for all other monoterpenes (Table S2). The emissions of the last set were treated as if they were emission of carene only. The median fraction of the emitted monoterpenes along the air-mass trajectories that were not α-pinene, β-pinene and limonene was 32 %.

LPJ-GUESS was run with the same meteorological data as used for determining the air-mass trajectories (GDAS (Rolph, 2016)) using 3-hourly data for 2005-2010, preceded by a spinup of 500 years to establish the vegetation and soil pools. Photosynthesis production and emissions of isoprene and monoterpenes were computed at the 3-hourly resolution of the GDAS data using air temperature and radiation, resulting in diurnal variations of the plants' transpirational demand and water stress. The maximum photosynthetic capacity along with water and leaf nitrogen content varied daily, following the daily averages of GDAS data. Land use was prescribed at the level of 2005 following Ahlström et al. (2012).

### 2.2.2 Primary particle emission

Primary particle emissions of wind-generated marine aerosol, as well as from ship and road traffic were included.

The primary marine aerosol production was estimated when the air-mass trajectories passed over ocean (determined by the land-use map from GLC2000) based on a parameterization from Mårtensson et al. (2003), with the use of wind-speed data from GDAS. The particles were assumed to be composed of NaCl and organic material based on the measurements and analysis of marine aerosol particles from Mace Head in Ireland during high biological activity (O'Dowd et al., 2004).

The emission of particles from ship and road traffic were estimated based on the $SO_2$ emission from ship and $NO_x$ emission from road traffic respectively, both retrieved from EMEP. For the ship emissions, a conversion factor of $8.33 \cdot 10^{14}$ particles/g($SO_2$) (Beecken et al., 2015) was used while a conversion factor of $2 \cdot 10^{14}$ particles/g($NO_2$) (Kristensson et al., 2004) was used for the road traffic emissions. Kristensson et al. (2004) also provided parameters for the size distribution of the traffic emissions. The size distribution of the particles from the ship emissions was based on a study done by Jonsson et al. (2011). The smallest particles (diameter less than or equal to 40 nm) were assumed to consist of 50 % H2SO4 and 50 % organic material. Particles larger than 40 nm were assumed to have a core of soot (black carbon) coated with a 5 nm thick layer of equal molar fractions of $H_2SO_4$ and organic material.

### 2.3 ADCHEM

ADCHEM can be used as a two, one or zero-dimensional model to simulate the aging of an air mass along a trajectory (Hermansson et al., 2014; Roldin et al., 2011a, 2011b). This section will focus on the modifications done to the model; for a detailed description of the model the reader is referred to Roldin et al. (2011b). In this study ADCHEM was used as a one-dimensional column model that solves the atmospheric diffusion equation in the vertical direction. The model included 20 vertical grid cells with a linear grid resolution of 100 m, extending up to 2000 m a.g.l. The vertical diffusion coefficient ($K_z$)



was calculated based on a slightly modified Grisogono-scheme (Jericevic et al., 2010) so that $K_z$ in Eq. (1) depends on the height above ground ($z$), the friction velocity ($u_*$) and the height of the atmospheric boundary layer ($H$):

$$K_z = Cu_* z \exp\left[-0.5\left(z/(0.21H)\right)^2\right] \tag{1}$$

where $C$ is an empirical constant estimated from large eddy simulation (LES) data. The cloud base was always assumed to lie above the model domain, i.e. no in-cloud aerosol processing was considered. Low-level clouds might have been present 34 % of the modeled times in the modeled domain on average, indicated by RH-values above 98 %.

The gas-phase chemistry was solved using the Kinetic PreProcessor (KPP) (Damian et al., 2002) with selected organic and inorganic reactions from The Master Chemical Mechanism (MCM) version 3.3 (Jenkin et al., 1997; Saunders et al., 2003) and with spectral irradiance modeled with the radiative transfer model described in Roldin et al. (2011b). Table S1 lists the gas-phase precursors included in the chemistry module. The two monoterpenes α-pinene and limonene that contain endocyclic double bonds were assumed to form HOMs initiated by their reaction with ozone. The HOM autoxidation mechanism was adopted from Ehn et al. (2014) and coupled to the MCMv3.3 mechanism. The HOM mechanism explicitly describes how the composition of the peroxy radicals ($RO_2$) formed from $O_3$ oxidation of monoterpenes evolves as a result of sequential steps of intramolecular H-shifts and $O_2$ additions (autoxidation) (Crounse et al., 2013). In this work in total 9 % of the first generation α-pinene + $O_3$ oxidized products were assumed to undergo autoxidation, while for limonene this fraction was 22 %. These numbers give upper limits for the molar yield of HOM formation from ozonolysis of α-pinene and limonene in our model. However, because of potential termination of the autoxidation mechanism with NO, $HO_2$ or $RO_2$ already after 1 or 2 H-shifts plus $O_2$ additions, not all autoxidation products become HOMs (O : C >= 0.7). For conditions with low NO concentrations (as was generally the case for the simulations in this work) the modeled HOM molar yield of formation was close to the measured molar yields of ~7 % (for α-pinene) and ~17 % (limonene) in the Jülich Plant Atmosphere Chamber (JPAC) (Ehn et al., 2014). These HOM yields are substantially higher than what was reported from flow tube experiments by Jokinen et al. (2015). One possible explanation to the different yields between these two studies is that the residence time in JPAC was substantially longer than in the flow tube. With longer residence time the autoxidation is allowed to run closer to completion, and for limonene specifically, there is potential to react twice with ozone. Thus, the yields reported by Ehn et al. (2014) most likely better resemble the HOM yields at low NO conditions in the atmosphere.

For β-pinene ozonolysis the autoxidation channel is minor (Ehn et al., 2014) and was not considered in the model. According to Ehn et al. (2014) and Jokinen et al. (2015) products from OH oxidation of α-pinene, limonene and β-pinene can also undergo autoxidation that leads to formation of HOMs. Jokinen et al. (2015) estimated that the molar yields of formation of HOMs from OH oxidation of α-pinene, limonene and β-pinene are 13 %, 27 % and 17 % of the molar yield of HOM formation from α-pinene + O3 reactions, respectively. Based on these results together with the molar yield of HOM formation from α-pinene ozonolysis from Ehn et al. (2014) we estimated and used an upper limit molar yield of HOM formation from OH oxidation of α-pinene, limonene and β-pinene of 1, 1.5 and 2.5 %, respectively. Figure S2 shows the





modeled median gas-phase concentration of the HOMs during all modeled NPF-events using different methods to estimate their vapor pressures (described below).

The aerosol dynamics module in ADCHEM considers new particle formation, Brownian coagulation, dry and wet deposition and condensation/evaporation. The changes in the particle number size distribution due to condensation, evaporation or

coagulation were modeled using a full-stationary size-grid (Jacobson, 2005) consisting of 100 size-bins between 1.5 nm and 2.5 µm in dry diameter.

The nucleation rate ($J_{1.5}$), see Eq. (2), was assumed to be a function of the concentration of sulfuric acid and a first generation oxidation product of the included monoterpenes denoted $ELVOC_{nucl}$, formed with a molar yield of $10^{-5}$ for each monoterpene that reacted with OH. The low molar yield was chosen in order to prevent $ELVOC_{nucl}$ to have a substantial

contribution to the modeled particle growth. This parameterization was recommended by Roldin et al. (2015), based on model simulations of NPF experiments with real plant emissions in JPAC. First generation oxidation products from reactions with $O_3$ were not included in $ELVOC_{nucl}$ since these tend to give too many new particles during the night (Roldin et al., 2015).

$$J_{1.5} = K_1[H_2SO_4][ELVOC_{nucl}] \tag{2}$$

where $K_1 = 2 \cdot 10^{-11}$ cm$^3$/s.

This value of $K_1$ was chosen in order for the model to give the approximately correct total particle number concentration if averaged over all model simulations. $K_1$ was kept constant for all modeled nucleation events.

As an alternative to Eq. 2 the model was also run with kinetic $H_2SO_4$ nucleation:

$$J_{1.5} = K_2[H_2SO_4][H_2SO_4] \tag{3}$$

where $K_2 = 2 \cdot 10^{-14}$ cm$^3$/s. In all model scenarios, the nucleation rate was determined by Eq. 2, if not otherwise noted.

Organic compounds with a pure liquid saturation vapor pressure ($p_0$) less than 0.01 Pa were included in the condensation mechanism, where $p_0$ was estimated with the group contribution method by Nannoolal et al. (2008) using the UManSysProp online system (Topping et al., 2016). The $p_0$ of the HOMs were estimated with the group contribution method SIMPOL (Pankow and Asher, 2008). The Nannoolal et al. (2008) method was not used for the HOMs because it was shown to

produce unrealistic estimates of vapor pressures for multifunctional HOMs containing hydroperoxide or peroxyacid groups (Kurtén et al., 2016). According to Kurtén et al. (2016) the SIMPOL method seems to be more robust and shows better agreement with the pure liquid vapor pressures of HOMs calculated with the detailed quantum-chemistry based continuum solvent model COSMO-RS (Conductor-like Screening Model for Real Solvents) (Eckert and Klamt, 2002) than the Nannoolal method. The SIMPOL method does however give substantially lower vapor pressures than COSMO-RS. Thus, a

sensitivity test was done where the vapor pressures of the HOMs calculated with SIMPOL were corrected based on the difference between the SIMPOL and COSMO-RS HOM vapor pressures reported by Kurtén et al. (2016) (Fig. S3). This yielded a correction factor of $10^{2.8 \cdot O:C - 0.1}$, where O : C is the oxygen-to-carbon ratio of the HOM monomers. For the HOM dimers we used a fixed correction factor of $10^4$.



The HOMs are probably very reactive in the particle phase and could therefore possibly be considered to be effectively non-volatile, despite their surprisingly high pure liquid saturation vapor pressures (Kurtén et al., 2016; Zhang et al., 2015). We evaluated the potential impact of irreversible reactive uptake of HOMs by performing simulations where the $p_0$ for the HOMs were set to zero, i.e. assuming that because of rapid irreversible reactions at the particle surface the HOM uptake is only limited by the collision rate between the HOMs and the particles.

ADCHEM includes a detailed particle-phase chemistry module, adopted from the Aerosol Dynamics gas- and particle phase chemistry model for laboratory CHAMber studies (ADCHAM) (Roldin et al., 2014). This module is used to calculate the particle equilibrium water content, the particle acidity, nitric acid and hydrochloric acid equilibrium vapor pressures for each particle size bin, and the non-ideal interactions between organic compounds, water and inorganic ions using the activity coefficient model AIOMFAC (Zuend et al., 2008, 2011). In this work, we did not simulate the specific interactions between the organic and inorganic compounds, but assumed a complete phase-separation of the inorganic and organic particle phase. We used AIOMFAC to calculate the equilibrium water content in the inorganic particle phase and the individual compound activity coefficients. The organic compound activity coefficients in the organic particle phase were not calculated in this work but were assumed to be unity (ideal solution). The equilibrium vapor pressures of the organic compounds over the particle surface were derived from $p_0$ using Raoult's law and correcting for the Kelvin effect, using a surface tension of 0.05 N m$^{-1}$ (Riipinen et al., 2010). The condensation, dissolution and evaporation of $NH_3$ and $HNO_3$ were calculated using the non-equilibrium growth scheme from Jacobson (2005). $H_2SO_4$ was treated as a non-volatile compound, with irreversible condensation.

ADCHEM can be combined with a kinetic multilayer model for particles (Roldin et al., 2014) where each particle consists of a surface-bulk layer and several bulk layers. In this study, the particles were either treated as liquid-like with no mass-transport limitations between the layers or as solid-like with no diffusion in the particle-phase. In the base-case simulations all particulate material except the core of the particles formed from soot particles were treated as liquid-like. The solid-like particles were represented with 3-layers (a monolayer thick surface layer of 0.7 nm and two bulk-layers). When the particles grow by condensation, material is moved from the surface layer into the first bulk-layer.

## 2.4 Initial conditions

The initial particle size distribution was assumed to be a typical distribution found in clean maritime air (Seinfeld and Pandis, 2006) where 90 % of the dry particle molar volume had the same chemical composition as the primary marine aerosols in Sect. 2.2.2 and the remaining dry volume consisted of ammonium sulfate.

The initial gas concentrations of $NO_x$, $SO_2$, $O_3$ and CO were retrieved from MACC (Monitoring Atmospheric Composition and Climate) reanalysis dataset (Inness et al., 2013) archived in the ECMWF data server.



**2.5 Sensitivity tests**

Sensitivity tests were done to investigate the impact of the selected NPF mechanism (Eq. 2 or Eq. 3) and how the growth of the particles was affected by the volatility of the HOMs and the SOA phase state of the particles. Table 1 lists the properties of the base-case simulation together with those of the sensitivity tests.

**3 Results and discussion**

This section presents the median characteristics of the modeled particle number concentration compared to the measured concentrations at Pallas. The results from the sensitivity tests of the model mentioned in section 2.5 will also be presented. First, however, model results from a typical day of observed new particle formation event are discussed.

Figure 2 shows the modeled (base-case simulation) and measured particle number size distribution at Pallas the 5[th] of July

2006. The model is able to reproduce the main features of the observed particle growth. At the beginning of the new particle formation event, around 09 UTC (11 local standard time), almost 90 % of the modeled particle volume in the nucleation mode consists of HOMs, the remaining volume largely consists of organic oxidation products from the MCMv3.3 chemistry scheme and sulfate (Fig. 3a). 9 hours later that day (Fig 3b) the particles originating from the NPF event form a new Aitken mode with a geometric mean diameter of ~50 nm. The volume fraction of VOC products from MCMv3.3 in the particle

phase is now slightly larger than at 09 UTC. This is partly because the Kelvin effect becomes insignificant when the particles have reached ~50 nm in diameter, which allows more SVOCs to dissolve in the organic aerosol particles.

**3.1 Median particle number size distribution**

In Fig. 4 a-d the observed and modeled (base-case scenario) median particle number size distributions for all chosen trajectories are presented together with their respective 25 and 75 percentiles. The newly formed particles reach the DMPS

detection limit size of 7 nm in diameter around noon local time (10 UTC) and have by early morning the day after produced particles around 80 nm, large enough to be able to act as CCN.

In the model simulation that included only organic chemistry from MCM v3.3 and no production of HOMs, the newly formed particles were not able to grow to observed sizes (Fig. 5 a-d). The gas-phase oxidation products without the HOM-chemistry do not reach low enough volatility to explain the observed growth rate and the particles are not able to grow to

CCN-sizes.

While the model base-case scenario is able to capture the main features of the observed particle growth, it generally (looking at the median of all 10 NPF-events) over-predicts the total number of particles larger than 7 nm in diameter at the beginning of the NPF (Fig. 6a). This might be caused by a too fast initial growth of the newly-formed particles (1.5 to 7 nm in diameter) or that the onset of the NPF-event happens too early in the model.

Two sensitivity tests were done to investigate the influence of the vapor pressure of the HOMs on the particle growth. When the HOMs were assumed to be non-volatile the modeled number of particles larger than 7 nm in diameter was very similar to





the base-case scenario (Fig. 6b). However; the simulation where the vapor pressures of the HOMs were corrected based on COSMO-RS, which resulted in higher vapor pressures of the HOMs, showed better agreement with the observations (Fig. 6c). When the HOM formation was excluded, the modeled NPF had only a minor influence on the total particle number concentration of particles larger than 7 nm in diameter (Fig. 6d).

The amount of particles larger than 50 nm in diameter during the evening and the day after the NPF event in all four model simulations mentioned above (liq-SIM HOM, liq-NV HOM, liq-COSMO HOM and liq-no HOM) is smaller than the observed particle number concentration (Fig. 7a-d, see also Fig. S4-S5 for number of particles above 30 and 80 nm in diameter). Most likely this is because the model underestimates the growth of the particles > 20 nm in diameter. The small contribution of LVOCs (saturation concentration below 0.3 µg m$^{-3}$) and SVOCs (saturation concentration above 0.3 µg m$^{-3}$)

to the particle growth (Fig. 8a-c) could be a reason why the model tends to underestimate the growth of particles larger than ~20 nm in diameter. Tröstl et al. (2016) showed that in order to explain the observed growth rates of particles in the full size range between ~1 to 30 nm in diameter, during an α-pinene ozonolysis experiment in the CERN CLOUD chamber, they needed to substantially increase the concentrations of SVOCs and LVOCs in their volatility basis set (VBS) model, compared to what was observed with an nitrate chemical ionization atmospheric pressure interface time of flight mass

spectrometer (nitrate-CI-APi-TOF). The motivation behind this VBS modification is that the nitrate-CI-APi-TOF likely underestimates the concentrations of HOMs in the SVOC and LVOC volatility range. Figure 8 shows the mean mass fraction of each compound type that contributes to the growth during all chosen NPF-events, from roughly the start time of the events (06 UTC) until the morning after (06 UTC). The growth of the particles is dominated by HOMs; the base-case simulation and the simulation with non-volatile HOMs both give HOM mass fractions of ~ 75 % on average. The simulation where the

vapor pressures of the HOMs are corrected with COSMO-RS predicts HOM mass fractions of ~ 50 % due to the higher vapor pressures of the HOMs. The fractions of total VOC-products from MCM in the particle phase (LVOC + SVOC) are ~ 10 % and ~ 20 % respectively.

Due to the dominance of HOMs, the O : C of the modeled SOA were substantially higher than typically reported (liq-SIM HOM: 0.99, liq-NV HOM: 0.98 and liq-COSMO HOM 0.93, compared to reported values from aerosol mass spectrometry

of 0.73 for aged low volatile SOA (Ng et al., 2010)). In a study by Zhang et al. (2015) they imply that particle-phase reaction can lower the high O : C of the HOMs (O : C > 0.7) to ratios they observe in the aerosol mass spectrometer. In our study, particle-phase reactions of HOMs were not modeled explicitly. The reason for the high O : C of the HOMs is that the autoxidation and formation of HOMs are relatively rapid processes which are not strongly hindered by the gas-to particle uptake of intermediate autoxidation products with lower O : C. Furthermore, the relatively low BVOC concentrations in the

atmosphere compared to most laboratory smog chamber experiments prevent substantial HOM dimer formation via RO$_2$ + RO$_2$ reactions according to the model. These reactions would lead to earlier termination of the autoxidation and formation of HOMs with lower O : C. One possible explanation to the high O : C ratio of the modeled SOA compared to atmospheric observations could be the lack of particle-phase reactions involving SVOCs with low O : C in the model, which would allow more SVOCs to partion to the particle phase. This, possibly together with the underestimated SVOC formation rates, can





also explain why the model underestimates the number of particles larger than 50 nm in diameter the day after the NPF events (Fig. 7), even though it seems to overestimate the initial growth (Fig. 6).

In Fig. 9, the influence of the phase state of the particles on the mean mass fraction of each compound type that contributes to the growth during all chosen new particle formation events is shown. Figure 9a presents the modeled results from the base-case simulation where the particles are assumed to be liquid and the vapor pressures of the HOMs are estimated with SIMPOL. Figure 9b shows results from the sensitivity test where the vapor pressures of the HOMs are the same as for the base-case but the particles are assumed to be solid. The differences between the simulations are minor. The most notable difference is that the fraction of nitrate is higher for particle sizes around 500 nm in diameter when the particles are assumed to be solid. The reason for this is that the solid surface layer, composed of low-volatility HOM SOA, traps the ammonium nitrate in the particle interior. The evaporation of ammonia and nitric acid will therefore be inhibited when the particles are solid as opposed to when they are liquid. The semi-volatile products (SVOCs) from the MCM-chemistry are not as much affected by the phase state of the particles as the nitrate. As opposed to the nitrate, the SVOCs are continuously replenished in the gas phase due to the continuous BVOC emissions over the forest. The result from this study implies that in environments with higher ammonia and $NO_x$ emission, the phase state of the particles could be an important factor to take into consideration. In the boreal environment of this study however, ammonium nitrate formation only contributes to a minor fraction of the secondary particle mass formation (Fig. S6) and does not contribute to the growth of the newly formed particles during the NPF events (Fig. 9).

The modeled volatility distribution of the SOA at Pallas is shown in Fig. 10 as the mean of all trajectories that were modeled (base-case scenario) at different times during the respective new particle formation event. At the beginning of the event the SOA mass is relatively small and almost completely dominated by HOM C10. As the particles grow, more high volatility material is able to condense. It can also be seen that the fraction of the HOM dimers (HOM C20) increases during nighttime as the NO is depleted. The shape of the VBS distribution in Fig.10 is in very good agreement with the fitted VBS distribution reported by Tröstl et al. (2016) (Extended Data Figure 5). They report a ELVOC:LVOC:SVOC ratio of 7:77:16. This can be compared to the average ELVOC:LVOC:SVOC ratio of 6.5:78.6:14.9 in Fig. 10. Figure S7 shows the volatility distribution modeled with the liq-COSMO HOM scenario.

To test the influence of the nucleation rate on particle growth, a sensitivity test was done where kinetic $H_2SO_4$ nucleation (Eq. 3) was used. On average, the kinetic sulfuric acid nucleation mechanism, as implemented in this work, caused more particles to form but the concentration of larger particles was fairly insensitive to the change in nucleation mechanism (Fig. S8 and S9).

# 4 Conclusions

During recent years the HOM-formation from endocyclic monoterpenes has been studied in laboratory and field environments (Ehn et al., 2014; Jokinen et al., 2015). In this study we evaluated the importance of this HOM formation in a



boreal environment and found that the model was able to capture the main features of the observed formation and growth rates during the studied NPF-events. The model could fully explain the activation and growth of new particles between 1.5 and ~20 nm in diameter. Between ~20 nm and 80 nm in diameter the model seems to underestimate the particle growth, even if the HOMs were assumed to be non-volatile. At the same time the model gives very high O : C of nearly 1 for the SOA.

Possible explanations to this could be that we did not consider particle phase oligomerization involving SVOCs in the model or that the model underestimates the SVOC formation rate from BVOCs. With more SVOCs and particle-phase oligomerization, mainly the growth of the larger particles (> 20 nm in diameter) would increase and the O : C decrease.

The modeled SOA mass formation was dominated by condensation of HOMs. However, the estimation of the vapor pressures of HOMs is very uncertain. A recent study by Kurtén et al. (2016) suggests that the vapor pressures might be

higher than previously thought and that the contribution of HOMs in the particle phase might be due to rapid reactions in the particle phase. We performed a sensitivity test where the vapor pressures of the HOMs were in line with values in Kurtén et al. (2016) and found that the model then seemed to explain the initial growth of the particles better than in the simulation with lower vapor pressures.

The growth of the particles was found to be independent on the phase state of the particles; the phase state might however be

of importance when the fraction of semi-volatile particulate matter is higher. In these cases, enrichment of low-volatility organic compounds at the particle surface might act as a protective shield against evaporation of SVOCs, ammonia and nitric acid.

### Acknowledgements

This work was carried out with the support by Nordic Center of Excellence programs CRAICC (Cryosphere- Atmosphere

Interactions in a Changing Arctic Climate) and eSTICC (eScience tools for investigating Climate Change in Northern High Latitudes), the European Union's Horizon 2020 research and innovation programme under grant agreement No 654109, the European Research Council (Grant 638703-COALA), the Swedish Strategic Research Program MERGE, Modeling the Regional and Global Earth System, the Lund Centre for studies of Carbon Cycle and Climate Interaction – LUCCI, the Swedish Research Council for Environment, Agricultural Sciences and Spatial Planning FORMAS (Project No. 214-2014-

1445) and by the Academy of Finland Center of Excellence program (project number 272041).

The authors would like to thank Fredrik Söderberg at the Centre for Environmental and Climate Research at Lund University for providing help to set up the model at the high performance computing cluster available at the center for scientific and technical computing at Lund University (Lunarc), and Finnish Meteorological Institute for providing the measurement data at Pallas.



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



**Table 1. Different assumptions for the different model scenarios tested in this study.**

| Model scenario | Phase-state | HOM vapor pressure method | Nucleation rate ($cm^3 s^{-1}$) |
|---|---|---|---|
| liq-SIM HOM (base-case) | liquid | SIMPOL | $J_{1.5} = 2 \cdot 10^{-11}[H_2SO_4][ELVOC_{nucl}]$ |
| liq-NV HOM | liquid | non-volatile | $J_{1.5} = 2 \cdot 10^{-11}[H_2SO_4][ELVOC_{nucl}]$ |
| liq-COSMO HOM | liquid | SIMPOL, corrected with COSMO-RS | $J_{1.5} = 2 \cdot 10^{-11}[H_2SO_4][ELVOC_{nucl}]$ |
| solid-NV HOM | solid | non-volatile | $J_{1.5} = 2 \cdot 10^{-11}[H_2SO_4][ELVOC_{nucl}]$ |
| solid-SIM HOM | solid | SIMPOL | $J_{1.5} = 2 \cdot 10^{-11}[H_2SO_4][ELVOC_{nucl}]$ |
| liq-no HOM | liquid | no HOMs included | $J_{1.5} = 2 \cdot 10^{-11}[H_2SO_4][ELVOC_{nucl}]$ |
| liq-kin nucl | liquid | non-volatile | $J_{1.5} = 2 \cdot 10^{-14}[H_2SO_4][H_2SO_4]$ |





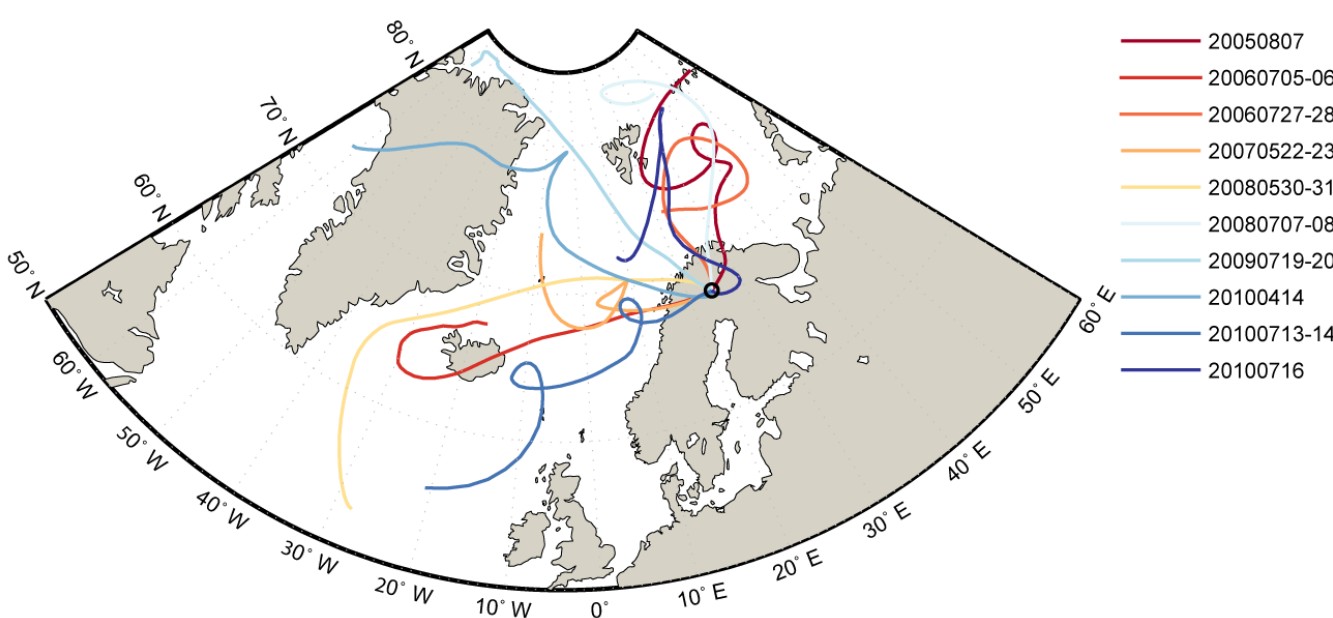

**Figure 1: Mean HYSPLIT trajectories of each new particle formation event, all ending at Pallas. The trajectories start 7 days backward in time before they reach the measurement station.**





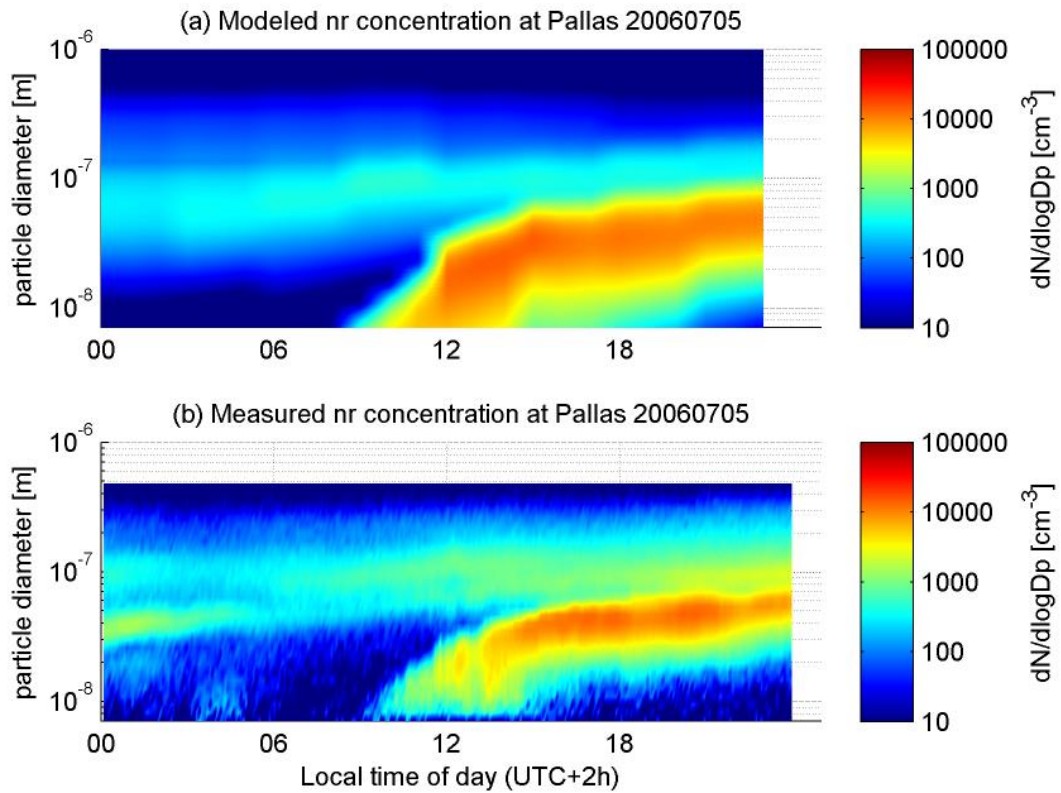

Figure 2. (a) Modeled and (b) measured number size distribution at Pallas the 5th of July 2006.



**Figure 3. Particle composition at (a) 09 and (b) 18 UTC at Pallas the 5$^{th}$ of July 2016. Solid lines are total particle volume concentration. The dashed lines are the modeled contributions of different compounds in the particle-phase.**

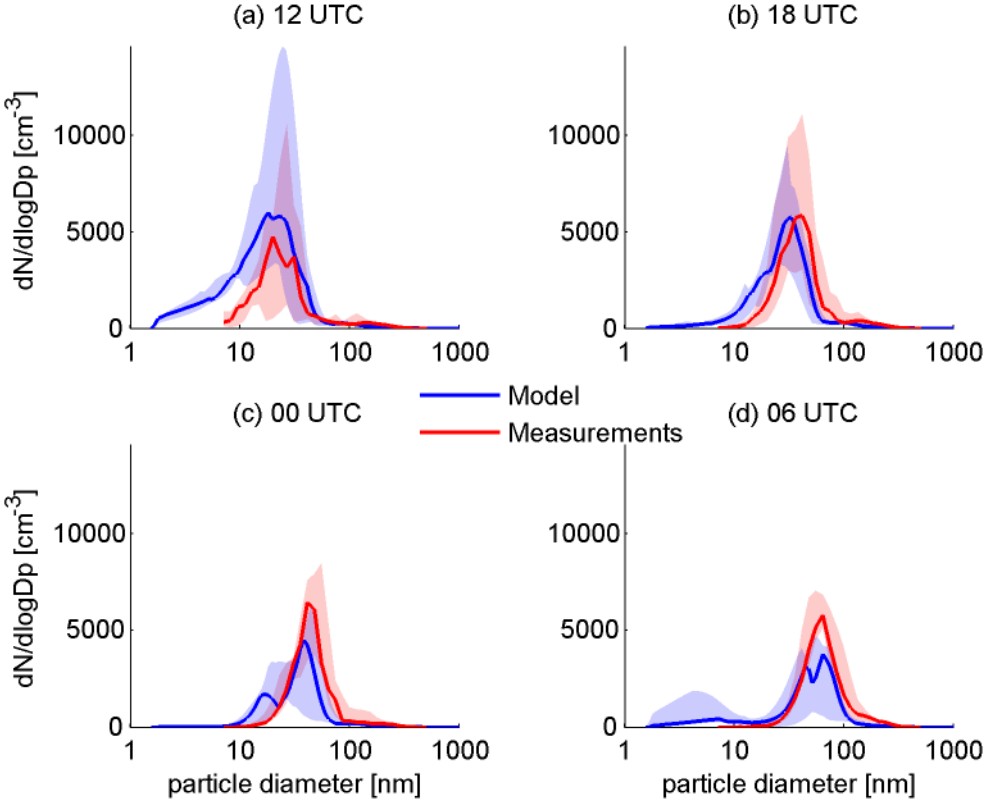

**Figure 4. The modeled particles are assumed to be liquid and the vapor pressures of the HOMs are estimated with SIMPOL. Measured (red lines) and modeled (blue lines) median number size distributions at (a) 12 and (b) 18 UTC the day of the new particle formation event and (c) 00 and (d) 06 UTC the following day. The shaded areas are the values that fall between the 25th and 75th percentiles.**





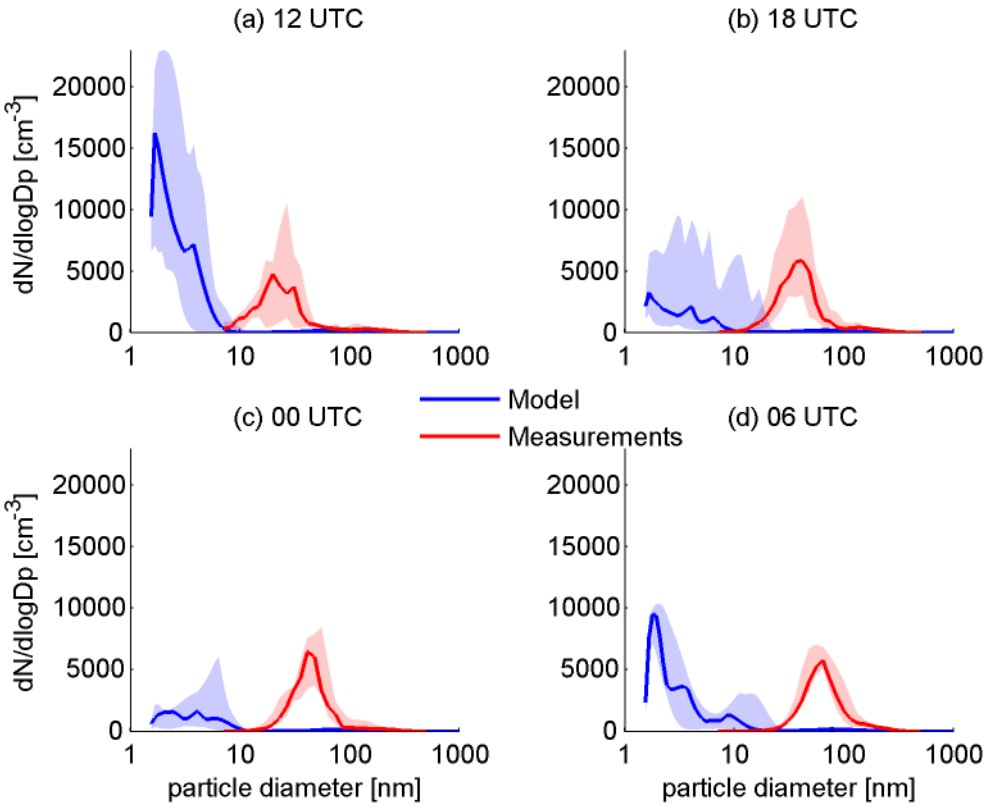

**Figure 5. The modeled particles are assumed to be liquid and the formation of HOMs is excluded. Measured (red lines) and modeled (blue lines) median number size distributions at (a) 12 and (b) 18 UTC the day of the new particle formation event and (c) 00 and (d) 06 UTC the following day. The shaded areas are the values that fall between the 25th and 75th percentiles.**





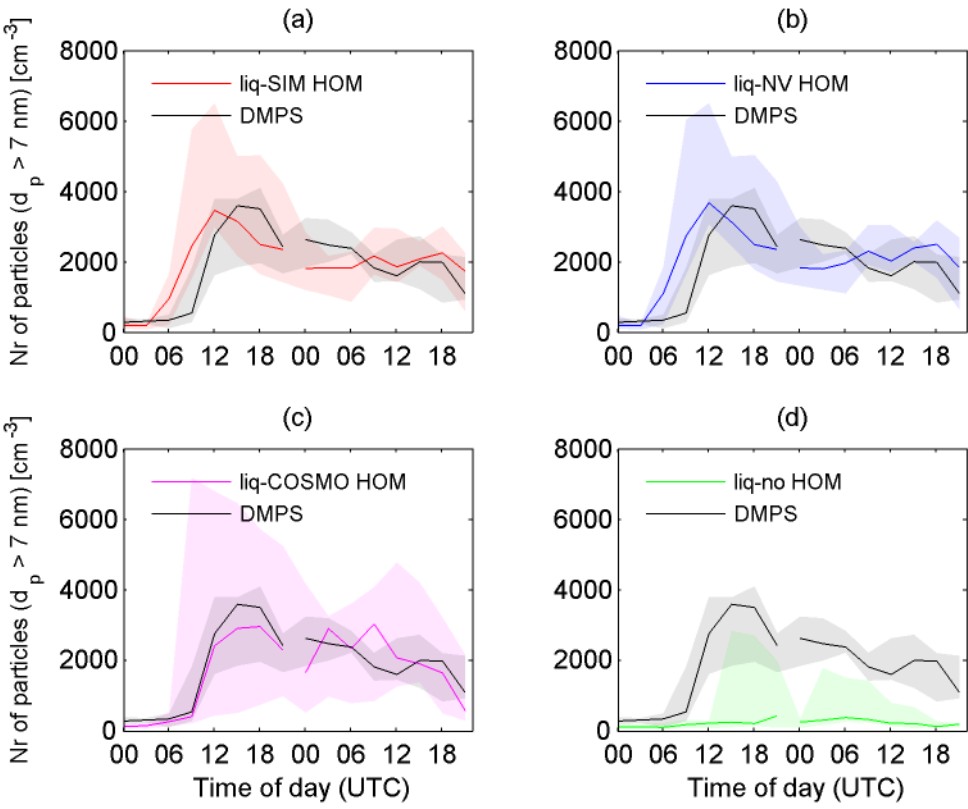

**Figure 6. Median number of particles above 7 nm of all chosen NPF-events at Pallas (from midnight at the day of the event to the evening the day after the start of the event) together with the 25 and 75 percentiles (shaded areas). The black lines are the median DMPS-data from Pallas. The colored lines in (a)-(c) are the modeled median number of particles above 7 nm, using different methods to estimate the vapor pressures of the HOMs (see table 1). In (d), HOMs are excluded.**



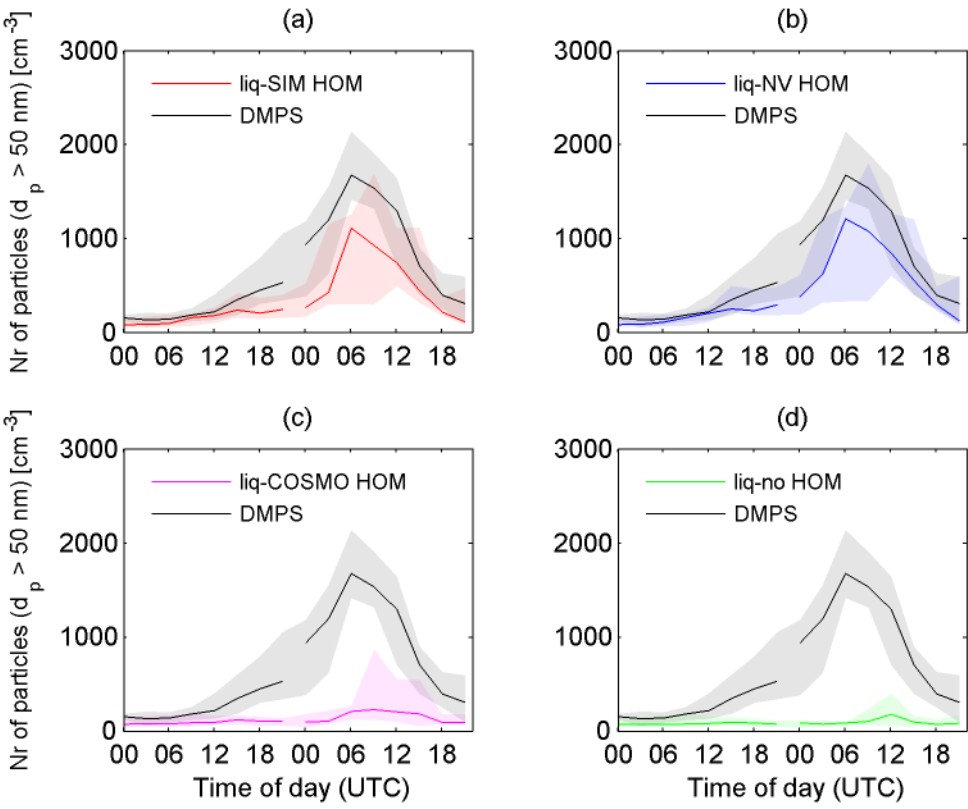

**Figure 7. Median number of particles above 50 nm of all chosen NPF-events at Pallas (from midnight at the day of the event to the evening the day after the start of the event) together with the 25 and 75 percentiles (shaded areas). The black lines are the median DMPS-data from Pallas. The colored lines in (a)-(c) are the modeled median number of particles above 50 nm, using different methods to estimate the vapor pressures of the HOMs (see table 1). In (d), HOMs are excluded.**





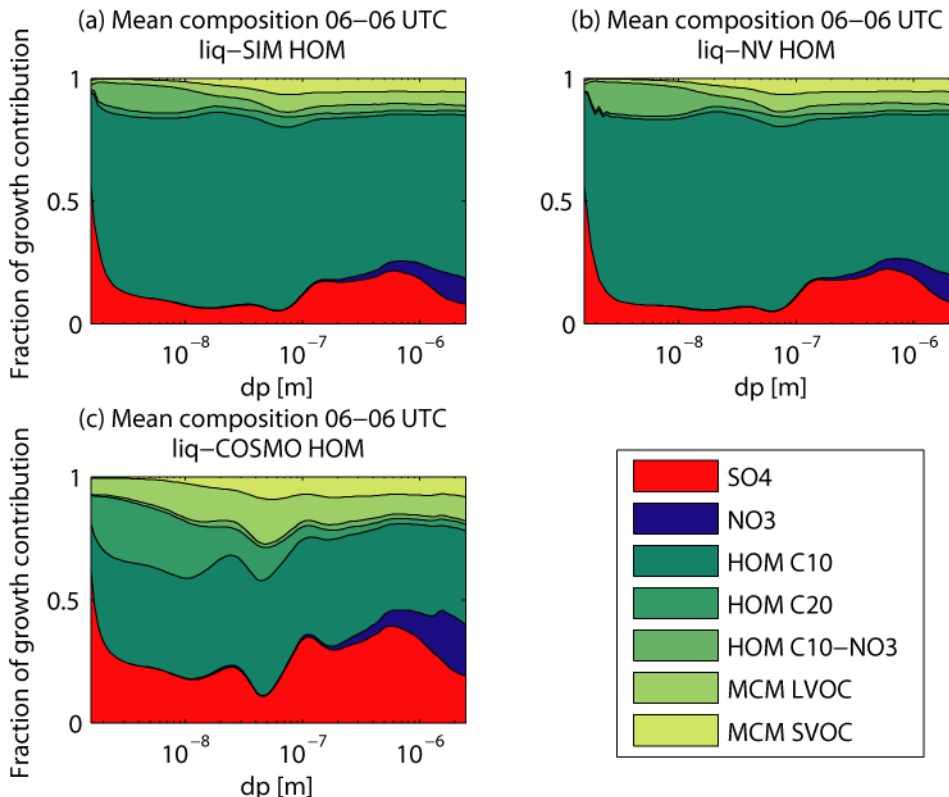

**Figure 8. Mean mass fractions of each compound type that contributes to the growth of the particles during all chosen new particle formation events (from 06 UTC the morning of the event to 06 UTC the following day). In (a) the particles are assumed to be liquid with vapor pressures of HOMs estimated with SIMPOL. In (b) the particles are assumed to be liquid and the vapor pressures of HOM non-volatile. In (c) the particles are assumed to be liquid with vapor pressures of HOMs estimated with SIMPOL but corrected for with COSMO-RS.**





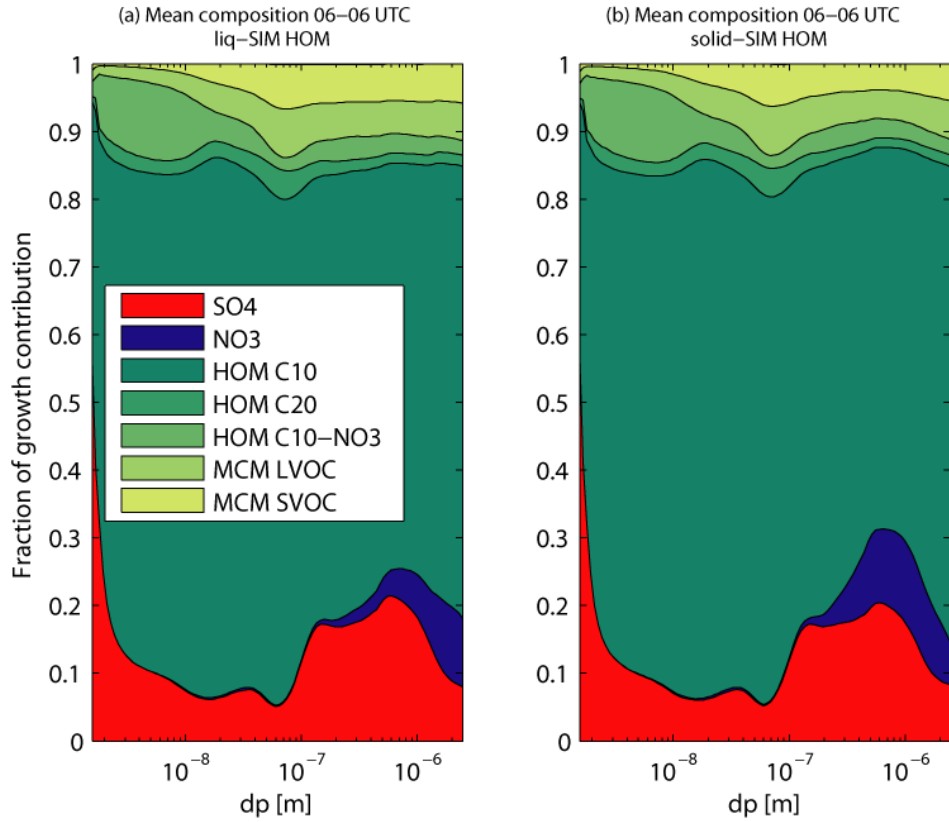

**Figure 9.** Mean mass fractions of the compound types that contribute to the growth of the particles during all chosen new particle formation events (from 06 UTC the morning of the event to 06 UTC the following day). In (a) the particles are assumed to be liquid with vapor pressures of HOMs estimated with SIMPOL. In (b) the particles are assumed to be solid with the same vapor pressure estimation.





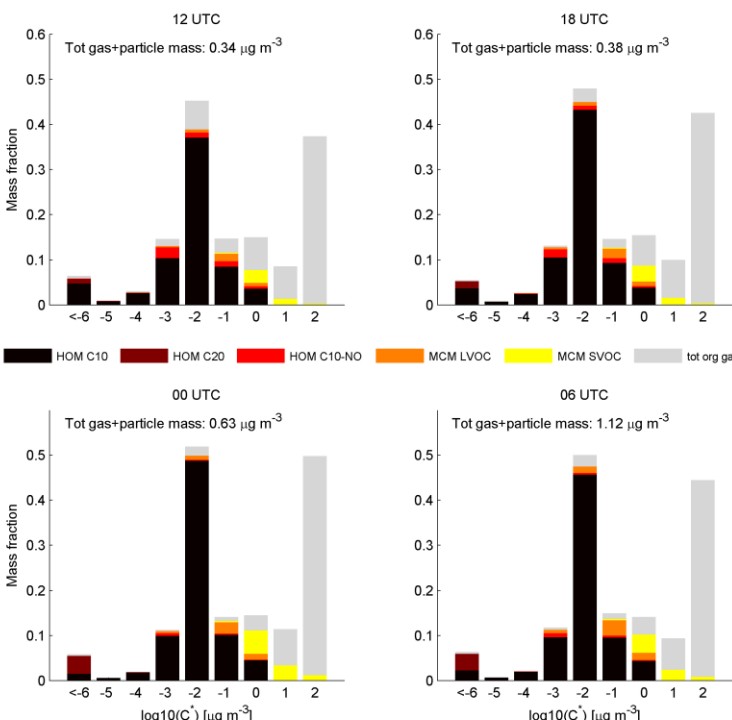

**Figure 10. Modeled mean volatility distribution of SOA-components at Pallas for different times ((a) 12 UTC, (b) 18 UTC, (c) 00 UTC and (d) 06 UTC) during new particle formation events. The gray bars are the sum of all oxidized organic compounds in the gas phase with C\* <= $10^2$ μg m$^{-3}$. The mass in each volatility bin is normalized to the total mass (gas and particle phase) of compounds with C\* <= 1 μg m$^{-3}$. The particles are assumed to be liquid and the vapor pressures of the HOMs are estimated with SIMPOL.**