# Peer review of "Modeling the role of highly oxidized multifunctional organic molecules for the growth of new particles over the boreal forest region"

_Atmospheric Chemistry and Physics, 2016_

## Referee Comment (RC1) · Anonymous Referee #1 · 4 Jan 2017

Öström et al. present a modeling study of growth dynamics and chemistry during new particle formation events at the Pallas boreal forest site in Northern Finland. They use an updated version of ADCHEM as a one-dimensional column trajectory model to simulate aerosol dynamics and chemistry of selected new particle formation events observed at the Pallas station between 2005 and 2010. The modeled particle number size distributions are compared to the measured size distributions, and the modeled contributions to particle growth of different chemical compounds and compound classes, in particular highly oxidized multifunctional organic compounds (HOMs), are discussed.

The manuscript is written in a clear and structured way, and covers a very interesting and timely topic, i.e. the role of HOMs in new particle formation and growth. This mod-

eling study is instructive and a useful contribution to the field. However, the title and the aim defined at the end of the Introduction section raise very high expectations, which are not entirely fulfilled in the presented manuscript. Therefore, I recommend considering the manuscript for publication after revisions taking into account the following comments:

1) The abstract is somewhat lengthy and should be shortened to focus on the key aspects of the manuscript. For example, the very first part of the abstract is quite general and could be more concise.

2) On p.3 line 17, "to constrain the contribution of HOMs to the activation and growth of new particles over the boreal forest region" is given as the aim of the study. This is a bold aim, and I don't find any results or conclusions that would constrain the contribution of HOMs to new particle growth over the boreal forest. Please revise the definition of the study aim(s). Also, can you really make a clear statement about the role of HOMs for the growth of new particles over the boreal forest region? I recommend rephrasing the manuscript title to better reflect the nature of the study - a comparison of modeled and measured particle number size distributions and model simulations of the chemical composition of new particles during growth.

3) p.5 line 5: What is the reasoning behind the treatment of all monoterpenes other than alpha-pinene, beta-pinene, and limonene as carene? Is this a common procedure?

4) p.6 lines 5/6: Clouds and in-cloud aerosol processing were not considered in the study. I think this is fine as a first approach, but given that 34 % of the modeled times may have been influenced by low-level clouds, the potential of aqueous-phase chemistry should be added to the Discussion section, both for the oxidation of organic compounds and H2SO4 formation (presented as gas phase reaction of SO2 and OH on p.4 line 27).

5) p.6 line 32: How exactly did you calculate the upper limit molar yield of HOM formation from OH oxidation? In line 29, you give molar yields of HOM formation from

OH oxidation of alpha-pinene, limonene, and beta-pinene as 13 %, 27 %, and 17 % of the molar yield of HOM formation from alpha-pinene ozonolysis. Then, I would assume that the molar yield of HOM formation from OH oxidation is highest for limonene, which is not the case according to the numbers given in line 32 (beta-pinene 2.5 % vs. limonene 1.5 %)?

6) p.8 lines 10-14: How sensitive are the results about the mass fractions of compound types contributing to particle growth given in Figures 8 and 9 to neglecting interactions between the organic and inorganic compounds, and to setting all organic compound activity coefficients equal to unity?

7) p.12 lines 1-7: Two main conclusions of the study are that the modeled particle size distributions show good agreement with the observations during initial growth up to 20 nm diameter, but underestimate particle growth in the diameter range from 20 - 80 nm. One plausible explanation given in the manuscript is that particle-phase oligomerization involving semi-volatile organic compounds has not been considered in the model. Such particle-phase reactions might increase particle growth and possibly reduce the very high O:C ratio of nearly 1 for the modeled SOA. How would this affect the modeled chemical composition and SOA volatility distribution presented in Figures 8 - 10? This is a very important finding, which should lead to follow-up studies. In my opinion, the statements that the model was able to "capture" the main features of the observed growth (p.1 line 24; p.2 line 1; p.9 line 26; p.12 line 1) should be carefully rephrased. Which physical and chemical features were explained by the model?

Technical comments:

p.3 line 1; p.11 line 32: The presence of gas-phase HOMs has been shown in more than the two studies given as a reference. Please add "e.g." before the given references.

p.3 line 14: Here, the reference should be given as (Roldin et al., 2011a), and modified throughout the manuscript accordingly.

p.5 line 24: Add superscripts in "H2SO4".

p.6 line 4: Give the value of C (= 0.39?) used in your study.

p.6 line 30: Add superscript in "O3".

p.10 line 14: Change to "... was observed with a nitrate chemical...".

p.17 line 19: Change to "Silver Spring".

p.30 Figure 10 and supplementary Figure S7: Explain the meaning of "HOM C10-NO" shown as red bars.

---

## Referee Comment (RC2) · Anonymous Referee #2 · 8 Jan 2017

The following is a review of "The role of highly oxidized multifunctional organic molecules for the growth of new particles over the boreal forest region" by Öström et al. This manuscript describes a modeling study of new particle formation and growth in which the performance of the model was assessed by comparing predicted particle number size distributions with those measured at the Pallas Station in the boreal forest in Northern Finland. It has long been recognized that improved model representations of the growth of nanometer-sized particles formed from nucleation are needed in order to adequately assess the potential role of new particle formation on atmospheric chemistry and climate. This manuscript can play an important role in addressing this need. However, I have some concerns about this manuscript that should be addressed

in order for me to consider this suitable for publication in Atmospheric Chemistry and Physics. These are listed below, in no particular order.

1. As the title suggests, this study focuses on the question of whether the partitioning of highly oxidized multifunctional organic molecules (HOMs) could account for all of the observed growth in the atmosphere above the boreal forest. The implementation of the 1D model (ADCHEM) used in this study was impressive in its ability to account for unique emissions along the air parcel that encounters the measurement site. However the model, as described on page 8, makes very simple assumptions regarding gas-particle partitioning of HOMs. Essentially, all HOMs are represented as undergoing non-reactive, reversible partitioning. Particle phase chemistry is assumed to be non-existent and HOMs that partition into the particle phase form ideal solutions. This simplification is recognized numerous times by the authors, but is so far from representing aerosol chemistry that I fear that the main conclusions from the study have been pre-determined by their modeling approach. I feel a more accurate title would refer specifically to reversible partitioning of low-volatility HOMs.

2. Model predictions of chemical composition are extremely important in this study, however no field measurements of particle chemistry are provided and I am certain that such data exist for the Pallas Station. For example, for many years FMI has participated in the Pallas Cloud Experiment (PaCE), where aerosol properties were measured along with cloud properties. Without some chemical composition data to compare modeling results to, it seems likely that the model could be getting the right answer for the wrong reasons. The only two reference to prior Pallas measurements was applied to a few sentences that described their DMPS and state that new particle formation occurs there, however there are numerous published studies to which the authors may refer (http://en.ilmatieteenlaitos.fi/pallas-publications). The only comparisons to measurements of particle composition refer to measurements at the CLOUD chamber, which mostly focus on processes for particle sizes smaller than those measured at Pallas and for which the authors make no attempt to define relevance by comparing the

chemical and environmental conditions of the two studies. I feel that the authors must provide some discussion of that which is known about Pallas aerosol, preferably co-incident measurements but even measurements performed during other time periods would still provide some insight. Even studies that have taken place at Hyytiälä, some of which have focused on characterizing the composition of nanometer-sized particles (ref: the published work of Johnston and Smith groups) would be useful.

3. There is much discussion in the introduction of the importance of aerosol formation and growth to CCN populations; however, a key component of this is understanding vertical transport of newly formed particles to parts of the atmosphere where they have the ability to affect cloud properties. While I understand that this manuscript is focusing on growth, ADCHEM was used in this study as a 1D vertical resolving model and can provide important insights into the potential role that these formation events may play on climate. I am sure that readers would be very interested to know if the events that were observed on the ground might potentially represent populations throughout the boundary layer. If the authors could comment on this in the manuscript, they would actually address the very same crucial research needs that they claim motivate their study.

4. If the SIMPOL model calculates that HOM vapor pressures are very low, then HOMs will partition to particles irrespective of diffusion limitations that may exist in particles. Therefore, one would not expect any changes in size-resolved chemistry in particles when comparing fast and slow particle phase mass transport in particles, and in fact this is borne out in Figure 9. What is the reader to learn from this exercise? Wouldn't it have been better to use the somewhat higher vapor pressures calculated using COSMO, especially if the authors felt that it better represented processes in the small diameter range?

5. Overall I felt that the Results and Discussion section of this manuscript provided very little analysis of the data. Rather, the discussion minimally knits together references to each of the 8 figures and sometimes just stated rather obvious attributes of

the figures while providing very little insight. For example, in reference to Figure 5 the authors state that the model run that did not include auto-oxidation mechanisms did not grow particles large enough to replicate the measured size distribution. This is clear from the figure. But there is no discussion of what the model DID predict, and this might be important when one considers environments in which auto-oxidation is curtailed by RO2-RO2 or RO2-NO chemistry. Another example: what is the importance of the distributions in Figure 10? They look nearly identical, except for small differences in the lowest volatility bins (which the authors attribute to dimerization). Why doesn't the volatility distribution evolve? Why compare the volatility distribution to Trostl et al.? Where the precursors and conditions (temperature, radiation, etc.) in that experiment identical to those at Pallas? Would it not be accurate to compare these distributions to those obtained at some of the SMEAR stations? Additionally, the authors state that lack of particle-phase chemistry of SVOC is likely to blame for many of the observed discrepancies between measured and modeled size distributions. As I say in item (1), it is of little surprise that ignoring LVOC and SVOC chemistry will lead to such problems. Studies like this modeling one can actually provide some insights into the types of processes that may actually explain observed growth rates. Even if the authors chose not to perform additional calculations, it would be useful for them to provide an assessment of the potential mechanisms that could help to address these discrepancies. Based on the list of co-authors, I believe that additional insights that would boost the quality of the analysis in this paper are indeed possible.

---

## Author Comment (AC1) · 17 Mar 2017

The comment was uploaded in the form of a supplement:
http://www.atmos-chem-phys-discuss.net/acp-2016-912/acp-2016-912-AC1-supplement.pdf

———————————————

---

## Author Response (AR1)

We thank the referees for the comprehensive and constructive comments on our manuscript. They definitely help use to improve the manuscript. Below you can find the answers to all comments and suggested modifications of the manuscript. All author comments are given in red text while the referee comments are given in *italic black text*. The suggested changes in the manuscript are given in *red italic*.

**Answers to referee 1:**

*Öström et al. present a modeling study of growth dynamics and chemistry during new particle formation events at the Pallas boreal forest site in Northern Finland. They use an updated version of ADCHEM as a one-dimensional column trajectory model to simulate aerosol dynamics and chemistry of selected new particle formation events observed at the Pallas station between 2005 and 2010. The modeled particle number size distributions are compared to the measured size distributions, and the modeled contributions to particle growth of different chemical compounds and compound classes, in particular highly oxidized multifunctional organic compounds (HOMs), are discussed. The manuscript is written in a clear and structured way, and covers a very interesting and timely topic, i.e. the role of HOMs in new particle formation and growth. This modeling study is instructive and a useful contribution to the field. However, the title and the aim defined at the end of the Introduction section raise very high expectations, which are not entirely fulfilled in the presented manuscript. Therefore, I recommend considering the manuscript for publication after revisions taking into account the following comments:*

*1) The abstract is somewhat lengthy and should be shortened to focus on the key aspects of the manuscript. For example, the very first part of the abstract is quite general and could be more concise.*

Yes, we agree with the reviewer. The abstract has been shortened and made more concise:

*2) On p.3 line 17, "to constrain the contribution of HOMs to the activation and growth of new particles over the boreal forest region" is given as the aim of the study. This is a bold aim, and I don't find any results or conclusions that would constrain the contribution of HOMs to new particle growth over the boreal forest. Please revise the definition of the study aim(s). Also, can you really make a clear statement about the role of HOMs for the growth of new particles over the boreal forest region? I recommend rephrasing the manuscript title to better reflect the nature of the study - a comparison of modeled and measured particle number size distributions and model simulations of the chemical composition of new particles during growth.*

We are aware of that we cannot prove that it actually are HOM formed via autoxidation that dominates the growth of the newly formed particles or that they are involved in the initial new particle formation mechanism. The only way this can be done is if we can develop reliable measurement techniques where the HOM species found in the gas-phase also can be detected in the particles and this way give a mass closure between what is found in the particles and in the gas-phase. Because of their presumable reactive nature this will most likely be very challenging. What we do in this study is to evaluate the potential role of HOM in the activation and growth of new particles over the boreal forest region. "*Constrain*" is probably a too strong wording. Our HOM mechanism and corresponding HOM formation yields for $\alpha$-pinene,

limonene and *β*-pinene are based on experiments from the Jülich Atmosphere Plant Chamber (Ehn *et al.,* 2014). We think that these are the best estimates of HOM formation from monoterpenes that we can use at present.

We have changed the phrase "to constrain the contribution of HOMs to the activation and growth of new particles over the boreal forest region"
To:
"to *evaluate* the *potential* contribution of HOMs to the activation and growth of new particles over the boreal forest region"

We have also changed the title to:
*Modelling the role of highly oxidized multifunctional organic molecules for the growth of new particles over the boreal forest region*

*3) p.5 line 5: What is the reasoning behind the treatment of all monoterpenes other than alpha-pinene, beta-pinene, and limonene as carene? Is this a common procedure?*

The reason why the choose to treat the 32 % of monoterpene emissions that were not alpha-pinene, beta-pinene or limonene as carene is that measurements in the boreal forest at the SMEAR II station in southern Finland show that carene together with α-pinene is the dominating VOC emitted from the Scots pine dominated forest (Pinus sylvestris  L.) see e.g. Hakola et al Biogeosciences, 3, 93–101, 2006 and Bäck *et al.*, Biogeosciences, 9, 689–702, 2012. Smolander et al., 2014 calculated the average monoterpene emissions from 40 Scots pine trees growing at SMEAR II to be approximately 43.7 % α-pinene and 39.6 % carene. At the hemiboreal SMEAR-Estonia site which as the boreal forest at Pallas is dominated by Norway Spruce Bourtsoukidis et al., 2014 measured that 25.8 % of the monoterpenes emitted from a Norway Spruce tree was carene while α-pinene made up 58.7 % of the total monoterpenen emissions. We will add a short motivation:

"The emissions of the last set were treated as if they were emission of carene only. The median fraction of the emitted monoterpenes along the air-mass trajectories that were not α-pinene, β-pinene and limonene was 32 %. *Carene was chosen to represent the generic set of monoterpenes because measurements on individual trees indicate that after α-pinene, carene is dominating the emissions from boreal forest composed predominantly of Scots pine (e.g. Bäck et al., 2012 and Smolander et al., 2014) or Norway spruce (Bourtsoukidis et al., 2014).*"

*4) p.6 lines 5/6: Clouds and in-cloud aerosol processing were not considered in the study. I think this is fine as a first approach, but given that 34 % of the modeled times may have been influenced by low-level clouds, the potential of aqueous-phase chemistry should be added to the Discussion section, both for the oxidation of organic compounds and H2SO4 formation (presented as gas phase reaction of SO2 and OH on p.4 line 27).*

Yes, we agree with the reviewer. We will add the following text to the result and discussion section:
*"It is likely that the model underestimates the sulfate mass in the accumulation*

*mode particles because we did not consider aerosol in-cloud processing and heterogeneous sulfate formation by oxidation of SO₂ in the cloud droplets. Also water-soluble organic compounds may be involved in heterogeneous reactions leading to additional SOA formation in the accumulation mode (e.g. Topping et al., 2013b). However, it is unlikely that this can explain why the model underestimates N₅₀ the day after the NPF events."*

Here $N_{50}$ refer to the number concentration of particles larger than 50 nm in diameter. This has also been stated in the updated result and discussion section.

*5) p.6 line 32: How exactly did you calculate the upper limit molar yield of HOM formation from OH oxidation? In line 29, you give molar yields of HOM formation from OH oxidation of alpha-pinene, limonene, and beta-pinene as 13 %, 27 %, and 17 % of the molar yield of HOM formation from alpha-pinene ozonolysis. Then, I would assume that the molar yield of HOM formation from OH oxidation is highest for limonene, which is not the case according to the numbers given in line 32 (beta-pinene 2.5 % vs. limonene 1.5 %)?*

Thank you for noticing this inconsistency. It is a misspelling from our side. It should be an upper molar yield of 1.5 % for $\beta$-pinene and a 2.5 % for limonene.
We have changed this sentence to:

"Based on these results together with the molar yield of HOM formation from α-pinene ozonolysis from Ehn et al. (2014) we estimated and used an upper limit molar yield of HOM formation from OH oxidation of α-pinene, *β-pinene* and *limonene* of 1, 1.5 and 2.5 %, respectively"

The exact upper limit molar yield values for OH oxidation was estimated as:
For $\alpha$-pinene: 9*0.13 = 1.17 (rounded to 1 %)
For $\beta$-pinene: 9*0.17=1.53 (rounded to 1.5 %)
For limonene: 9*0.27=2.43 (rounded to 2.5 %)

*6) p.8 lines 10-14: How sensitive are the results about the mass fractions of compound types contributing to particle growth given in Figures 8 and 9 to neglecting interactions between the organic and inorganic compounds, and to setting all organic compound activity coefficients equal to unity?*
This is a good question. However, we feel that it is beyond the scope of this article to answer it. We have previously evaluated the effect of considering non-ideal mixing of SOA from α-pinene ozonolysis and ammonium using AIOMFAC, where we calculated the activity coefficients of the organic compounds (Roldin *et al.,* 2014). According to these tests the modeled SOA formation was almost identical with or without explicit treatment of the interactions between inorganic and organic compounds. The model results was much more sensitive to which functional group contribution method that was used to estimate the vapour pressures of the organic compounds. This is consistent with the conclusions drawn by Topping et al, Faraday Discuss., 165, 273–288, 2013. We have added a motivation to why we did not consider organic and inorganic interactions and to why we decided to use unity organic compound activity coefficients for this study before line 10-14 on page 8:

*"Topping et al., 2013 concluded that the uncertainties in modelled SOA formation is far greater because of uncertainties in the organic compound pure liquid saturation*

*vapour pressures than the omission of phase separation between organic and inorganic compounds. In line with this, we have previously shown that while the modelled SOA formation during α-pinene ozonolysis experiments is relatively sensitive to the choice of pure liquid saturation vapour estimation method, it is relative insensitive to the omission of non-ideal interactions between the condensable organic compounds and between the organic compounds and ammonium (Roldin et al., 2014). In Kurtén et at. (2016) we computed the activity coefficients of 16 different HOM in a water-insoluble organic matter phase using the COSMOTherm software (Eckert and Klamt, 2014) and found that the activity coefficients varied between 0.59 and 2.01. Thus,* in this work we did not simulate the specific interactions between the organic and inorganic compounds, but assumed a complete phase-separation of the inorganic and organic particle phase. We used AIOMFAC to calculate the equilibrium water content in the inorganic particle phase and the individual compound activity coefficients. The organic compound activity coefficients in the organic particle phase were assumed to be unity (ideal solution)."

*7) p.12 lines 1-7: Two main conclusions of the study are that the modeled particle size distributions show good agreement with the observations during initial growth up to 20 nm diameter, but underestimate particle growth in the diameter range from 20 - 80 nm.One plausible explanation given in the manuscript is that particle-phase oligomerization involving semi-volatile organic compounds has not been considered in the model. Such particle-phase reactions might increase particle growth and possibly reduce the very high O:C ratio of nearly 1 for the modeled SOA. How would this affect the modeled chemical composition and SOA volatility distribution presented in Figures 8 - 10? This is a very important finding, which should lead to follow-up studies. In my opinion, the statements that the model was able to "capture" the main features of the observed growth (p.1 line 24; p.2 line 1; p.9 line 26; p.12 line 1) should be carefully rephrased. Which physical and chemical features were explained by the model?*

We agree with the reviewer that this should be studied further in follow-up studies. This is something we are working on. With particle-phase oligomerization we expect that the contribution from SVOCs to the SOA mass would increase. The average SOA O:C ratio would most likely decrease because SVOCs generally have lower O:C ratio than HOM. The SOA mass with very low volatility $C* < 10^{-3} \mu g\ m^{-3}$ would probably increase and concentrations of the SVOCs in the gas-phase would be lower than expected from pure equilibrium gas-particle phase partitioning theory. The total SOA mass would most likely increase but the growth of the newly formed particles could potentially be supressed if the condensation sink onto larger particles increases.
Yes we agree with the reviewer that the very vague statements like "the model was able to capture the main features of the observed growth" should be replaced with a more concrete descriptions of what we figures actually shows.

On p. 1 line 24 we have reformulated the sentence:
"While the model seems to capture the growth of the newly formed particles between 1.5 and ~ 20 nm in diameter, it underestimated the particle growth between ~20 and 80 nm in diameter."
To:
*"In the model the newly formed particles with an initial diameter of 1.5 nm reach a diameter of 7 nm about 2 hours earlier than what is typically observed at the*

*station. This is an indication that the model tends to overestimate the initial particle growth. On the other hand, the modeled particle growth to CCN size ranges (>50 nm in diameter) seems to be underestimated because the increase in the concentration of particles above 50 nm in diameter typically occurs several hours later compared to the observations."*

We have removed the sentence on p. 2, line 1.

The result and discussion section has been partly rewritten based on the suggestions from both referees. Some vague and not precise statements have been removed or reformulated.

On p. 12, line 1-3 we have replaced the sentences:
"In this study we evaluated the importance of this HOM formation in a boreal environment and found that the model was able to capture the main features of the observed formation and growth rates during the studied NPF-events. The model could fully explain the activation and growth of new particles between 1.5 and ~20 nm in diameter."

with:

*"In this study we evaluated the importance of HOM formation from monoterpene autoxidation in a boreal environment. The modelled HOM formation rate is high enough to give sufficient condensable vapours to explain or even slightly overestimate the growth of the newly formed particles between 1.5 nm to ~20 nm in diameter, if most of the formed HOMs are LVOCs or ELVOCs."*

As last sentence in the abstract we have added:
*"Future studies should evaluate how heterogeneous reactions involving semi-volatility HOMs and other less oxidized organic compounds can influence the SOA composition and size dependent particle growth."*

*And to the conclusions we have added:*
*"We suggest that future studies should follow up on how heterogeneous reactions involving HOMs and other SVOCs influence the particle number size distribution evolution and the aerosol chemical composition during new particle formation events."*

Technical comments:
*p.3 line 1; p.11 line 32: The presence of gas-phase HOMs has been shown in more than the two studies given as a reference. Please add "e.g." before the given references.*
Thank you. We have added e.g.

*p.3 line 14: Here, the reference should be given as (Roldin et al., 2011a), and modified throughout the manuscript accordingly.*

Yes you are correct. We have change the order of the two Roldin et al., publications from 2011 in the reference list and refer to Roldin et al., 2011a instead of Roldin et al., 2011b.

*p.5 line 24: Add superscripts in "H2SO4".*
Done

*p.6 line 4: Give the value of C (= 0.39?) used in your study.*
Thank you. We have done this. C is equal to 0.34.

*p.6 line 30: Add superscript in "O3".*
Done

*p.10 line 14: Change to "... was observed with a nitrate chemical...".*
Thank you, we have replaced *an* with *a*.

*p.17 line 19: Change to "Silver Spring".*
Done

*p.30 Figure 10 and supplementary Figure S7: Explain the meaning of "HOM C10-NO" shown as red bars.*

Thank you for noticing this. It should be HOM $C10-NO_3$ that denotes HOM species with 10 carbon atoms and one nitrate functional group. We separated the organic nitrate HOM species from the other HOM monomers. We will explain this in the figure texts.

**Answers to referee 2:**

The following is a review of "The role of highly oxidized multifunctional organic molecules for the growth of new particles over the boreal forest region" by Öström et al. This manuscript describes a modeling study of new particle formation and growth in which the performance of the model was assessed by comparing predicted particle number size distributions with those measured at the Pallas Station in the boreal forest in Northern Finland. It has long been recognized that improved model representations of the growth of nanometer-sized particles formed from nucleation are needed in order to adequately assess the potential role of new particle formation on atmospheric chemistry and climate. This manuscript can play an important role in addressing this need. However, I have some concerns about this manuscript that should be addressed in order for me to consider this suitable for publication in Atmospheric Chemistry and Physics. These are listed below, in no particular order.

1. As the title suggests, this study focuses on the question of whether the partitioning of highly oxidized multifunctional organic molecules (HOMs) could account for all of the observed growth in the atmosphere above the boreal forest. The implementation of the 1D model (ADCHEM) used in this study was impressive in its ability to account for unique emissions along the air parcel that encounters the measurement site. However the model, as described on page 8, makes very simple assumptions regarding gas-particle partitioning of HOMs. Essentially, all HOMs are represented as undergoing non-reactive, reversible partitioning. Particle phase chemistry is assumed to be non-existent and HOMs that partition into the particle phase form ideal solutions. This simplification is recognized numerous times by the authors, but is so

far from representing aerosol chemistry that I fear that the main conclusions from the study have been pre-determined by their modeling approach. I feel a more accurate title would refer specifically to reversible partitioning of low-volatility HOMs.

We are aware of that particle phase reactions e.g. oligomerization do occur in aerosol particles. However, there are very few atmospheric chemistry transport models that explicitly take particle phase chemistry into account. We mention that such processes could explain part of the discrepancies between the modeled and measured particle growth rates but to take such processes into account in a realistic manner is not an easy task and we feel is beyond the scope of this study. We are working on this topic and hope to present such model results in a follow-up study. Since most of the HOM species in the present study are LVOC and ELVOCs, at least when using the SIMPOL functional group contribution method, particle phase processes will have little influence on the particle growth caused by HOMs. In a way the test simulations with non-volatile HOM species, on a particle growth perspective, take into account the extreme case where HOM instantaneously react and form non-volatile oligomers. However, for SVOCs we do believe that these processes are very important to consider and we point out this in the discussion as one of our main conclusions.

Yes, as stated we do not explicitly calculate organic compounds activity coefficients in the condensed phase. We are not aware of any other atmospheric chemistry transport model that does this explicitly either. As we also replied to referee 1 we have previously evaluated the effect of considering non-ideal mixing of SOA from α-pinene ozonolysis and ammonium using AIOMFAC, where we calculated the activity coefficients of the organic compounds (Roldin *et al.,* 2014). According to these tests the modelled SOA formation was almost identical with or without explicit treatment of the interactions between inorganic and organic compounds. The model results was much more sensitive to which functional group contribution method that was used to estimate the vapour pressures of the organic compounds. This is consistent with the conclusions drawn by Topping et al, Faraday Discuss., 165, 273–288, 2013. We have added a motivation to why we did not consider organic and inorganic interactions and to why we decided to use unity organic compound activity coefficients for this study before line 10-14 on page 8:

*"Topping et al., 2013 concluded that the uncertainties in modelled SOA formation is far greater because of uncertainties in the organic compound pure liquid saturation vapour pressures than the omission of phase separation between organic and inorganic compounds. In line with this, we have previously shown that while the modelled SOA formation during α-pinene ozonolysis experiments is relatively sensitive to the choice of pure liquid saturation vapour estimation method, it is relative insensitive to the omission of non-ideal interactions between the condensable organic compounds and between the organic compounds and ammonium (Roldin et al., 2014). In Kurtén et at. (2016) we computed the activity coefficients of 16 different HOM in a water-insoluble organic matter phase using the COSMOTherm software (Eckert and Klamt, 2014) and found that the activity coefficients varied between 0.59 and 2.01. Thus*, in this work we did not simulate the specific interactions between the organic and inorganic compounds, but assumed a complete phase-separation of the inorganic and organic particle phase. We used AIOMFAC to calculate the equilibrium water content in the inorganic particle phase and the individual compound activity coefficients. The organic compound activity coefficients in the organic particle phase were not

calculated in this work but were assumed to be unity (ideal solution)"

2. Model predictions of chemical composition are extremely important in this study, however no field measurements of particle chemistry are provided and I am certain that such data exist for the Pallas Station. For example, for many years FMI has participated in the Pallas Cloud Experiment (PaCE), where aerosol properties were measured along with cloud properties. Without some chemical composition data to compare modeling results to, it seems likely that the model could be getting the right answer for the wrong reasons. The only two reference to prior Pallas measurements was applied to a few sentences that described their DMPS and state that new particle formation occurs there, however there are numerous published studies to which the authors may refer (http://en.ilmatieteenlaitos.fi/pallas-publications). The only comparisons to measurements of particle composition refer to measurements at the CLOUD chamber, which mostly focus on processes for particle sizes smaller than those measured at Pallas and for which the authors make no attempt to define relevance by comparing the chemical and environmental conditions of the two studies. I feel that the authors must provide some discussion of that which is known about Pallas aerosol, preferably coincident measurements but even measurements performed during other time periods would still provide some insight. Even studies that have taken place at Hyytiälä, some of which have focused on characterizing the composition of nanometer-sized particles (ref: the published work of Johnston and Smith groups) would be useful.

We agree with the reviewer that we should compare the modeled particle chemical composition with results from observations. Unfortunately there are no coincident measurements of sub-micron aerosol chemical composition (e.g. using AMS) for any of the 10 selected new particle formation events that we study with our model. But we will compare our modeled chemical composition with the average chemical compositions of sub-micron aerosol particles originating from marine and Arctic air masses measured during the second and third Pallas Cloud Experiments (Kivekäs et al., BER, 2009 and Jaatinen et al., BER 2014). We will add the following section to the result and discussion section of the manuscript:

*"According to Kivekäs et al. (2009) the average detectable inorganic aerosol mass fraction (nitrate, ammonia and sulfate) was 23 %, and the remaining 77 % was organics for aerosol particles originating from marine air masses during the second Pallas Cloud Experiment conducted between 16th of September and 6th of October, 2005. During the third Pallas Cloud Experiment (21st of September 2009 to 3rd of October 2009), when the air masses were originating from Northern Atlantic Ocean and the Arctic, the AMS measurements together with black carbon measurements with a Multi-Angle Absorption Photometer (MAAP) gave an average composition of 47 % organics, 26 % sulfate, 13 % ammonia, 8 % nitrate and 6 % black carbon (Jaatinen et al., 2014). However, during the only strong new particle formation and growth event occurring during this campaign more than 70 % of the particles mass was composed of organics (Figure 2c in Jaatinen et al., 2014). Because of the generally very low mass concentrations (< 1 μg m$^{-3}$) during the second and third Pallas Cloud Experiment no reliable size resolved chemical composition could be derived from the AMS measurements. However, Jaatinen et al., (2014) compared the aerosol hygroscopicity parameter, κ, derived using the non-size resolved AMS chemical compositions with the size resolved κ derived with an HT-DMA and an CCN counter.*

*According to this closure the AMS κ was generally above 0.2 and substantially higher than the κ values derived with HT-DMA and a CCN counter. For particles with diameters between 15 and 75 nm the κ values were in the range between 0.05 and 0.08 based on the HT-DMA and CCN counter measurements. Jaatinen et al., (2014) also conclude that this is likely because the newly formed particles are mainly composed of organic compounds. In our base case simulation the secondary aerosol particle mass is also strongly dominated by organic compounds with an average mass fraction of 85 %, and with the remaining inorganic secondary aerosol mass fraction mainly being composed of sulfate (Fig 7). Thus, the ratio between the modeled total organic mass and the inorganic secondary aerosol mass (nitrate, ammonia and sulfate) is somewhat larger than reported by Kivekäs et al. (2009) and substantially larger than the average values from Jaatinen et al., (2014). However, both AMS measurement campaigns were performed during the autumn when the BVOC emissions from the boreal forest generally are relatively low, while our modeled cases mainly are from the late spring and summer months when the BVOC emissions generally are higher because of higher temperatures and photosynthetic active radiation (e.g. Schurgers et al., 2009a). Additionally, we have only focused on days with strong new particle formation and consecutive particle growth. Jaatinen et al., (2014) conclude that particular during these days the sub-micron particles are likely mainly composted of secondary organic material. It is likely that the model underestimates the sulfate mass in the accumulation mode particles because we did not consider aerosol in-cloud processing and heterogeneous sulfate formation by oxidation of $SO_2$ in the cloud droplets. Also water-soluble organic compounds may be involved in heterogeneous reactions leading to additional SOA formation in the accumulation mode (e.g. Topping et al., 2013b). However, it is unlikely that this can explain why the model underestimates $N_{50}$ the day after the NPF events."*

3. There is much discussion in the introduction of the importance of aerosol formation and growth to CCN populations; however, a key component of this is understanding vertical transport of newly formed particles to parts of the atmosphere where they have the ability to affect cloud properties. While I understand that this manuscript is focusing on growth, ADCHEM was used in this study as a 1D vertical resolving model and can provide important insights into the potential role that these formation events may play on climate. I am sure that readers would be very interested to know if the events that were observed on the ground might potentially represent populations throughout the boundary layer. If the authors could comment on this in the manuscript, they would actually address the very same crucial research needs that they claim motivate their study.

Thank you for this very good suggestion. We will add a figure illustrating the modeled median vertical profiles at Pallas of the modeled total particles number concentration of particle with diameters >7 nm and >50 nm at 12 UTC the day of the NPF events and 12 UTC the day after the NPF events and the following text:

*"Figure 6 shows the modeled median vertical concentration profiles of $N_7$ and $N_{50}$ at the Pallas field station at 12 UTC the days of the NPF events and at 12 UTC the day after the NPF events. $N_7$ and $N_{50}$ is elevated in the whole boundary layer to an altitude of ~800 m because of the previous day NPF events. Above the typical maximum boundary layer height of ~ 800 m $N_7$ decreases steeply from > 1000 cm$^{-3}$ to < 10 cm$^{-3}$ above 1600 m. Thus, according to these model results NPF events in*

*the sub-Arctic forest region can be an important source of CCN in the whole planetary boundary layer. Further, the observed $N_7$ and $N_{50}$ at the ground can give reasonable accurate estimates of $N_7$ and $N_{50}$ in the whole boundary layer but do not reflect the concentrations above the boundary layer, either during the NPF events or the day after the events."*

[Figure]

*Figure 6. Modeled median vertical profiles of the particle number concentrations of particles larger than > 7 nm in diameter ($N_7$) and > 50 nm in diameter ($N_{50}$), respectively. Model results are shown both from the first day during the NPF events at 12 UTC and the second day after the NPF events at 12 UTC. Shown are also the observed median particle number concentrations at the surface.*

4. If the SIMPOL model calculates that HOM vapor pressures are very low, then HOMs will partition to particles irrespective of diffusion limitations that may exist in particles. Therefore, one would not expect any changes in size-resolved chemistry in particles when comparing fast and slow particle phase mass transport in particles, and in fact this is borne out in Figure 9. What is the reader to learn from this exercise? Wouldn't it have been better to use the somewhat higher vapor pressures calculated

using COSMO, especially if the authors felt that it better represented processes in the small diameter range?

Yes, it would also be good to run the model with the COSMO vapor pressures too, and with solid like particles. As the referee correctly points out, if the growth is dominated by SVOCs then the phase state will be more important for the results. But since the model substantially underestimates the growth beyond 7 nm in diameter when we use the vapor pressures estimated based on COSMO, we decided to not do these model runs in the present work. In future studies we plan to run the model with the COSMO vapor pressures and including heterogeneous reactions that transform condensable SVOCs to ELVOCs in the particle phase and thereby enhance the growth. But this is not straightforward to implement in the model, and cannot be included in the present work. Then we also plan to test the effect of the SOA phase state.

We suggest that we put Figure 9 as Fig S12 in the supplement and rewrite the discussion about the impact of phase state on page 11, L3-L17 to:

"*We also evaluated the impact of the SOA phase by running the model as the base case model run but with solid-like SOA particles instead of liquid. The differences between the base case model runs and these simulations are minor (Fig. S11 in the supplement). One of the reasons for this is that the main fraction of the SOA is formed by condensation of LVOCs (Fig. 8). If a dominating fraction of the SOA instead would be SVOCs, the SOA phase state would most likely have a larger impact on the model results (see e.g. Zaveri et al., 2014). The most notable difference, in our model results is that the fraction of nitrate is higher for particle sizes around 500 nm in diameter when the particles are assumed to be solid. The reason for this is that the solid surface layer, composed of low-volatility HOM SOA, traps the ammonium nitrate in the particle interior. The evaporation of ammonia and nitric acid will therefore be inhibited when the particles are solid as opposed to when they are liquid. The SVOCs from the MCM-chemistry are not as much affected by the phase state of the particles as the ammonium nitrate. One likely reason for this is that, opposed to the ammonium nitrate, the SVOCs are continuously replenished in the gas phase due to the continuous BVOC emissions over the forest. The result from this study implies that in environments with higher ammonia and $NO_x$ emission or during conditions when the SOA formation mainly is driven by condensation of SVOCs, the phase state of the particles could be an important factor to take into consideration. However, in the boreal environment of this study, at least the ammonium nitrate formation generally only contributes to a minor fraction of the secondary particle mass formation (e.g. Jaatinen et al., 2014 and Fig. 7) and does not contribute to the growth of the newly formed particles during the NPF events (Fig. S12).*"

We will also change the sentence on L28-30 in the abstract from:
"According to the model the phase state of the SOA (assumed either liquid or amorphous solid) had an insignificant effect on the evolution of the particle number size distribution during the NPF events."
To:
"*In the model simulations where condensation of low-volatility and extremely low-volatility HOMs explains most of the SOA formation, the phase state of the SOA (assumed either liquid or amorphous solid) had an insignificant impact on the*

*evolution of the particle number size distributions.”*

5. Overall I felt that the Results and Discussion section of this manuscript provided very little analysis of the data. Rather, the discussion minimally knits together references to each of the 8 figures and sometimes just stated rather obvious attributes of the figures while providing very little insight. For example, in reference to Figure 5 the authors state that the model run that did not include auto-oxidation mechanisms did not grow particles large enough to replicate the measured size distribution. This is clear from the figure. But there is no discussion of what the model DID predict, and this might be important when one considers environments in which auto-oxidation is curtailed by RO2-RO2 or RO2-NO chemistry.

We agree with the referee that part of the results and discussion section can be improved. We suggest that we move Fig 3, 5, 9 to the supplement because we realize that they do not add any added value apart from what can be described shortly with a few sentences of text. We will modify Figure 8 and only include Fig 8a in the manuscript and put the complete figure Fig 8a-c in the supplement. We will also only include Fig 10c in the manuscript and put the complete Fig 10a-d in the supplement. Additionally, we would also like to include the figure with the vertical concentration profiles to the manuscript (see the answer to point 1).

We changed the text that describe the results in Fig 5 from:

“In the model simulation that included only organic chemistry from MCM v3.3 and no production of HOMs, the newly formed particles were not able to grow to observed sizes (Fig. 5 a-d). The gas-phase oxidation products without the HOM chemistry do not reach low enough volatility to explain the observed growth rate and the particles are not able to grow to CCN-sizes.”

To:
*“When the HOM formation was excluded, the modeled NPF had only a minor influence on $N_7$ (Fig. 4d), because most of the newly formed particles were not able to grow to observable sizes (Fig. S5). Thus, in more polluted environments where the autoxidation is terminated by $RO_2 + RO_2$ or $RO_2 + NO$ reactions before the oxidation products become HOM, the particle growth may be suppressed.”*

The text describing the results in Fig 6-10 has been completely rewritten.

Another example: what is the importance of the distributions in Figure 10? They look nearly identical, except for small differences in the lowest volatility bins (which the authors attribute to dimerization). Why doesn't the volatility distribution evolve? Why compare the volatility distribution to Trostl et al.? Where the precursors and conditions (temperature, radiation, etc.) in that experiment identical to those at Pallas? Would it not be accurate to compare these distributions to those obtained at some of the SMEAR stations?

The VBS distributions do evolve but the formation rates of SOA in the very clean atmosphere in the sub-arctic region is relatively slow. Further, the BVOC emissions along the modeled air mass trajectories that governing the SOA formation is relatively similar and thus the formed SOA composition is relatively similar over time. As we

clearly state in the manuscript we do not consider particle phase reactions that would cause the SOA VBS to change over time.

The conditions in the CLOUD experiments in CERN were not identical to the atmospheric conditions in our study but are similar in the way that the SOA formation was completely dominated by condensation of oxidation products formed from monoterpenes. In the cloud experiment it was only $\alpha$-pinene and in our model simulations $\alpha$-pinene account for approximately 40 % of the total monoterpene emissions. The temperature in the CLOUD chamber experiment was 278 K. This temperature is within $\pm$ 10 K of the temperature at Pallas for the selected NPF events. Approximately 60 % of the $\alpha$-pinene in the cloud chamber was oxidized by $O_3$ and the remaining 40 % by OH. In the model simulations the fraction of $\alpha$-pinene that is oxidized by $O_3$ varies from up to 90 % during the night to a minimum of about 30 % around noon.

We will add the following two sentences to the manuscript:

*"Although the experiments in Tröstl et al. (2016) do not fully represent the conditions in our atmospheric study, the SOA formation is in both cases dominated by ozonolysis and OH oxidation of monoterpenes. Thus, we think it is relevant to compare our modeled SOA volatility distribution with theirs."*

Yes, optimally it would also be good to compare the model VBS distributions with VBS distributions derived from field observations at a SMEAR station or similar. We are aware of studies using thermodenuder techniques to study the volatility of ambient aerosol particles but to derive VBS distribution from these results is not straight forward and rely on several assumptions (see e.g. Karnezi et al., Atmos. Meas. Tech., 7, 2953–2965, 2014). Usually the atmospheric aerosol particles are not only composed of SOA and the size resolved chemical information is usually not available with high enough temporal resolution. Hong et al. Atmos. Chem. Phys., 14, 4733–4748, 2014 did derive thermograms for ambient aerosol particles measured at the SMEAR II station that possible cloud be used to estimate VBS distributions but they did not do this. If the referee is aware of reliable VBS distribution parameterizations of ambient SOA sampled in Boreal forest environment we would be happy to compare our modeled results to those. However, we have not found such data.

Additionally, the authors state that lack of particle-phase chemistry of SVOC is likely to blame for many of the observed discrepancies between measured and modeled size distributions. As I say in item (1), it is of little surprise that ignoring LVOC and SVOC chemistry will lead to such problems. Studies like this modeling one can actually provide some insights into the types of processes that may actually explain observed growth rates. Even if the authors chose not to perform additional calculations, it would be useful for them to provide an assessment of the potential mechanisms that could help to address these discrepancies. Based on the list of co-authors, I believe that additional insights that would boost the quality of the analysis in this paper are indeed possible.

We do not agree with the referee that we ignore LVOC and SVOC chemistry. We do use a very detailed gas-phase chemistry mechanism that is state of the art within atmospheric chemistry transport modeling. We do not consider heterogeneous reactions involving LVOCs and SVOCs in the present study since the type and

reaction rates of these reactions are very uncertain and it is a very complex task to include such reactions in an accurate and computational feasible manner. Some of the authors of this study have some previous experience in simulating how heterogeneous reactions potentially can influence the SOA formation during smog chamber experiments (e.g. Roldin et al., ACP 2014 and 2015) but to include these mechanism in an accurate manner in the present study would require both substantial more man-power and computational resources. Still, it is our ambition that we in the future will try to do this. But not in the present study. Still, we do not ignore the effect that such reactions may cause since we discuss the potential effect that these reactions may have on the SOA formation and size resolved particle growth in the context of our model and measurement comparison. This way we believe that our study in indeed shed light on the processes " that may actually explain observed growth rates". We have chosen to not go into details concerning the exact heterogeneous reactions that could contribute to additional SOA formation involving SVOCs and LVOCs but instead more generally discuss the potential impact that such reactions may lead to. We believe that studies that intent to constrain the importance of specific heterogeneous mechanisms should start with well-controlled laboratory experiments with less complex SOA mixtures than what is the case in this atmosphere study.

We have added the following sentence to the end of the abstract:
*"Future studies should evaluate how heterogeneous reactions involving semi-volatility HOMs and other less oxidized organic compounds can influence the SOA composition and size dependent particle growth."*

And in the conclusion section we will write:

*"We suggest that future studies should follow up on how heterogeneous reactions involving HOMs and other SVOCs influence the particle number size distribution evolution and the aerosol chemical composition during new particle formation events."*